# Training on Test Data with Bayesian Adaptation for Covariate Shift

**Aurick Zhou, Sergey Levine**
Department of Electrical Engineering and Computer Sciences
University of California, Berkeley
`{aurick,svlevine}@berkeley.edu`

## Abstract

When faced with distribution shift at test time, deep neural networks often make inaccurate predictions with unreliable uncertainty estimates. While improving the robustness of neural networks is one promising approach to mitigate this issue, an appealing alternate to robustifying networks against all possible test-time shifts is to instead directly adapt them to unlabeled inputs from the particular distribution shift we encounter at test time. However, this poses a challenging question: in the standard Bayesian model for supervised learning, unlabeled inputs are conditionally independent of model parameters when the labels are unobserved, so what can unlabeled data tell us about the model parameters at test-time? In this paper, we derive a Bayesian model that provides for a well-defined relationship between unlabeled inputs under distributional shift and model parameters, and show how approximate inference in this model can be instantiated with a simple regularized entropy minimization procedure at test-time. We evaluate our method on a variety of distribution shifts for image classification, including image corruptions, natural distribution shifts, and domain adaptation settings, and show that our method improves both accuracy and uncertainty estimation.

## 1   Introduction

Modern deep learning methods can provide high accuracy in settings where the model is evaluated on data from the same distribution as the training set, but accuracy often degrades severely when there is a mismatch between the training and test distributions [Hendrycks and Dietterich, 2019, Taori et al., 2020]. In safety-critical settings, effectively deploying machine learning models requires not only high accuracy, but also requires the model to reliably quantify uncertainty in its predictions in order to assess risk and potentially abstain from making dangerous, unreliable predictions. Reliably estimating uncertainty is especially important in settings with distribution shift where inaccurate predictions are more prevalent, but the reliability of uncertainty estimates often also degrades along with the accuracy as the shifts become more severe [Ovadia et al., 2019]. In real-world applications, distribution mismatch at test time is often inevitable, thus necessitating methods that can robustly handle distribution shifts, both in terms of retaining high accuracy but also in providing meaningful uncertainty estimates.

As it can be difficult to train a single model to be robust to all potential distribution shifts we might encounter at test time, we can instead robustify models by allowing for *adaptation* at test time, where we finetune the network on unlabeled inputs from the shifted target distribution, thus allowing the model to specialize in the particular shift it encounters. Since test-time distribution shifts often cannot be anticipated during training time, we restrict the adaptation procedure to operate without any further access to the original training data. Prior work on test-time adaptation [Wang et al., 2020a, Sun et al., 2019b] focused on improving accuracy, and found that simple objectives like entropy minimization capable of providing substantial improvements under distribution shift [Wang et al.,

35th Conference on Neural Information Processing Systems (NeurIPS 2021).

2020a]. However, these prior works do not consider uncertainty estimation, and an objective like entropy minimization can quickly make predictions overly confident and less calibrated, leaving us without reliable uncertainty estimates and thus unable to quantify risks when making predictions. Our goal in this paper is to design a test-time adaptation procedure that can not only improve predictive accuracy under distribution shift, but also provide reliable uncertainty estimates.

While a number of prior papers have proposed various heuristic methods for test-time adaptation [Wang et al., 2020a, Sun et al., 2019a], it remains unclear what precisely unlabeled test data under covariate shift can actually tell us about the optimal classifier. In this work, we take a Bayesian approach to this question, and explicitly formulate a Bayesian model that describes how unlabeled test data from a different domain can be related to the classifier parameters. Such a model requires introducing an additional explicit assumption, as the classifier parameters are conditionally independent of unlabeled data in the standard model for discriminative classification [Seeger, 2000]. The additional assumption we introduce intuitively states that the data generation process at test-time, though distinct from the one at training time (hence, under covariate shift) is still more likely to produce inputs that have a single unambiguous labeling, even if that labeling is not known. We argue that this assumption is reasonable in practice, and leads to an appealing graphical model where approximate inference corresponds to a Bayesian extension of entropy minimization.

We propose a practical test-time adaptation strategy, Bayesian Adapatation for Covariate Shift (BACS), which approximates Bayesian inference in this model and outperforms prior adaptive methods both in terms of increasing accuracy and improving calibration under distribution shift. Our adaptation strategy is simple to implement, requires minimal changes to standard training procedures, and outperforms prior test-time adaptation techniques on a variety of benchmarks for robustness to distribution shifts.

## 2 Bayesian Adaptation for Covariate Shift

In this section, we will devise a probabilistic graphical model that describes how unlabeled data in a new test domain can inform our posterior about the model, and then describe a practical deep learning algorithm that can instantiate this model in a system that enables test-time adaptation. We will begin by reviewing standard probabilistic models for supervised and discuss why such models are unable to utilize unlabeled data. We then discuss a probabilistic model proposed by Seeger [2000] that does incorporate unlabeled data in a semisupervised learning (SSL) setting, and propose an extension to the model to account for distribution shift. Finally, we discuss the challenges in performing exact inference in our proposed model and describe the approximations we introduce in order to derive a tractable inference procedure suitable for test-time adaptation.

### 2.1 Probabilistic Model for Covariate Shift

In the standard Bayesian model for supervised learning (Figure 1), we assume that the inputs $\mathbf{X}$ are sampled i.i.d. from a generative model with parameters $\phi$, while the corresponding labels are sampled from a conditional distribution $p(Y|\mathbf{X}, \theta)$ parameterized by $\theta$. The parameters $\theta$ and $\phi$ are themselves random variables, sampled independently from prior distributions $p(\phi)$ and $p(\theta)$. We observe only a dataset $\mathcal{D} = \{\mathbf{X}_i, Y_i\}_{i=1}^n$, and then perform inference over the parameters $\theta$ using Bayes rule:

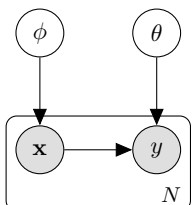

$$p(\theta|\mathcal{D}) \propto p(\theta) \prod_{i=1}^{n} p(Y_i|\mathbf{X}_i, \theta). \tag{1}$$

To make predictions on a new input $\mathbf{x}_{n+1}$, we marginalize over the posterior distribution of the classifier parameters $\theta$ to obtain the predictive distribution

Figure 1: Probabilistic model for standard supervised learning, observing $N$ labeled data points.

$$p(Y_{n+1}|\mathbf{X}_{n+1}, \mathcal{D}) = \int p(Y_{n+1}|\mathbf{X}_{n+1}, \theta)p(\theta|\mathcal{D}) \, d\theta.$$

Note that observing the (unlabeled) test input $\mathbf{x}_{n+1}$ (or more generally, any number of test inputs) does not affect the posterior distribution of parameters $\theta$, and so within this probabilistic model, there is no benefit to observing multiple unlabeled datapoints before making predictions.

**Inference for semi-supervised learning**. By treating the parameters $\theta, \phi$ as a-priori independent, the standard model assumes there is no relationship between the input distribution and the process that generates labels, leading to the inability to utilize unlabeled data in inferring $\theta$. To utilize unlabeled data, we can introduce assumptions about the relationship between the labeling process and input distribution using a model of the form in Figure 2 [Seeger, 2000].

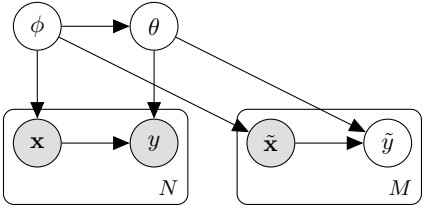

Figure 2: Model for SSL [Seeger, 2000], observing $N$ labeled pairs and $M$ unlabeled inputs, both generated from the same distribution. Observing unlabeled inputs $\tilde{\mathbf{x}}$ provides information about the input distribution $\phi$, and thus about $\theta$.

The assumption we use has a simple intuitive interpretation: we assume that the inputs that will be shown to our classifier have an unambiguous and clear labeling. This assumption is reasonable in many cases: if the classifier discriminates between automobiles and bicycles, it is reasonable that it will be presented with images that contain either an automobile or bicycle. Of course, this assumption is not necessarily true in all settings, but this intuition often agrees with how discriminatively trained models are actually used. This simple intuition can be formalized in terms of a prior belief about the *conditional entropy* of labels conditioned on the inputs.

Similar to Grandvalet and Bengio [2004], we can encode this belief using the additional factor

$$p(\theta|\phi) \propto \mu(\theta)\exp(-\alpha H_{\theta,\phi}(Y|\mathbf{X})) \tag{2}$$

$$= \mu(\theta)\exp\left(\alpha\mathbb{E}_{\mathbf{X}\sim p(X|\phi),Y\sim p(Y|X,\theta)}[\log p(Y|\mathbf{X},\theta)]\right), \tag{3}$$

where $\mu(\theta)$ is a prior over the parameters $\theta$ that is agnostic of the input distribution parameter $\phi$.

Now, observing unlabeled test inputs $\mathcal{U} = \{\tilde{\mathbf{x}}_1, \ldots, \tilde{\mathbf{x}}_m\}$ provides information about the parameter $\phi$ governing the input distribution, which then allows us to update our belief over the learned parameters $\theta$ through Equation 2 and thus allows inference to utilize unlabeled data within a Bayesian framework.

**Extension to covariate shift**. The previous probabilistic model for SSL assumes all inputs were drawn from the same distribution (given by $\phi$). However, our goal is use unlabeled data to adapt our model to a *different* test distribution, so we extend the model to incorporate *covariate shift*.

We now assume we have two input-generating distributions; $\phi$ specifies the input distribution for our labeled training set, and $\tilde{\phi}$ specifies the shifted distribution of unlabeled test inputs which we aim to adapt to. Under the assumption of covariate shift, the same classifier $\theta$ is used to generate labels in both the train and test domains, leading to the model in Figure 3.

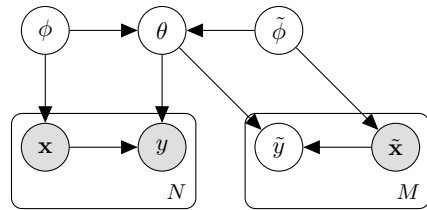

Figure 3: Our proposed probabilistic model for adaptation for covariate shift, observing a training set with $N$ labeled pairs and $M$ unlabeled inputs from a shifted test distribution.

We argue that our prior belief of low aleatoric uncertainty is reasonable for *both* the distribution induced by $\phi$ and the one induced by $\tilde{\phi}$; even if there is some distribution shift from the training set, we can still often expect the test data we are shown to have unambiguous labels. We can then incorporate both $\phi$ and $\tilde{\phi}$ into our belief over $\theta$ with the factor

$$p(\theta|\phi,\tilde{\phi}) \propto \mu(\theta)\exp(-\alpha H_{\theta,\phi}(Y|\mathbf{X}))\exp(-\tilde{\alpha}H_{\theta,\tilde{\phi}}(Y|\mathbf{X})), \tag{4}$$

with the two scalar hyperparameters $\alpha, \tilde{\alpha}$ controlling how to weight the entropies from each distribution with the likelihoods of the labeled training set. While we can extend this model to include more labeled or unlabeled input distributions, we focus on simply having one training distribution and one test distribution as it is the most relevant for our problem setting of adapting to a particular distribution shift at test time.

## 2.2 Approximations for Tractable Inference

Performing inference over $\theta$ in the above model can be challenging in our test-time adaptation setting for several reasons. First of all, inference over $\theta$ would also require performing inference over the

parameters of the generative models $\phi$ and $\tilde{\phi}$ to evaluate likelihoods in Equation 4. This is difficult in practice for two reasons: First, the inputs $x$ might be high-dimensional and difficult to model, as in the case of images. Second, the amount of unlabeled data might be fairly small, and insufficient to accurately estimate the parameters $\tilde{\phi}$ if we employ a highly expressive generative model. As such, we would much prefer to avoid explict generative modeling and to only perform inference over the discriminative model parameters $\theta$ instead.

Another issue is that performing inference would require access to both the labeled training data and the unlabeled test data at the same time, while our test-time adaptation setting assumes that we can no longer look at the training set when it is time to adapt to the test distribution. We will discuss how to address these issues and describe our method, Bayesian Adaptation for Covariate Shift (BACS), which provides a practical instantiation of inference in this Bayesian model in a computationally tractable test-time adaptation setting.

**Plug-in approximation with empirical Bayes**. We first propose to avoid explicit generative modeling by using with a plug-in empirical Bayes approach. Empirical Bayes is a common approximation for hierarchical Bayesian models that simply uses a point estimate of $\phi$ rather than marginalizing over $\phi$'s (similarly for $\tilde{\phi}$), which would reduce the computation to estimating only a single generative model for each input distribution given by parameters $\phi^*$ and $\tilde{\phi}^*$. To eliminate the need to train the parameters $\phi^*$ of a generative model of the inputs altogether, we note that $p(\theta|\phi, \tilde{\phi})$ only depends on $\phi$ and $\tilde{\phi}$ through the input distributions $p(x|\phi), p(\tilde{x}|\tilde{\phi})$. We can then approximate Equation 4 by plugging in the empirical distributions of $x$ and $\tilde{x}$ in place of $p(x|\phi^*), p(\tilde{x}|\tilde{\phi}^*)$, resulting in

$$p(\theta|\hat{\phi}, \tilde{\phi}) \propto \mu(\theta) \exp\left(-\frac{\alpha}{n} \sum_{i=1}^{n} H(Y_i|\mathbf{x}_i, \theta)\right) \exp\left(-\frac{\tilde{\alpha}}{m} \sum_{i=1}^{m} H(Y_i|\tilde{\mathbf{x}}_i, \theta)\right).$$

Now, given a labeled training set $\mathcal{D}$ and unlabeled test points $\mathcal{U} = \{\tilde{\mathbf{x}}_i\}_{i=1}^{m}$, the new posterior distribution over parameters now has log probabilities (up to an additive normalizing constant)

$$\log p(\theta|\mathcal{D}, \mathcal{U}) = \log \mu(\theta) + \sum_{i=1}^{n} \log p(y_i|\mathbf{x}_i, \theta) - \frac{\alpha}{n} \sum_{i=1}^{n} H(Y|\mathbf{x}_i, \theta) - \frac{\tilde{\alpha}}{m} \sum_{j=1}^{m} H(Y|\tilde{\mathbf{x}}_j, \theta). \quad (5)$$

For convenience, we simply set $\alpha = 0$, as additionally minimizing entropy on the labeled training set is unnecessary when we are already maximizing the likelihood of the given labels with highly expressive models. Now, to infer $\theta$ given the observed datasets $\mathcal{D}$ and $\mathcal{U}$, the simplified log-density is

$$\log p(\theta|\mathcal{D}, \mathcal{U}) = \log \mu(\theta) + \sum_{i=1}^{n} \log p(y_i|\mathbf{x}_i, \theta) - \frac{\tilde{\alpha}}{m} \sum_{j=1}^{m} H(Y|\tilde{\mathbf{x}}_j, \theta). \quad (6)$$

Computing the maximum-a-posteriori (MAP) solution of this model corresponds to simply optimizing model parameters $\theta$ that on the supervised objective on the labeled training set in addition a minimum entropy regularizer on the unlabeled input. However, we should not necessarily expect the MAP solution to provide reasonable uncertainty estimates, as the learned model is being encouraged to make confident predictions on the test inputs, and so any single model will likely provide overconfident predictions. Marginalizing the predictions over the posterior distribution over parameters is thus essential to recovering meaningful uncertainty estimates, as the different models, though each being individually confident, can still express uncertainty through their combined predictions when the models predict different labels.

**Approximate inference for test-time adaptation**. We now discuss how to perform inference in the above model in a way suitable for *test-time adaptation*, where we adapt to the test data without any further access to the original training set. To enable this, we propose to learn an approximate posterior

$$q(\theta) \approx p(\theta|\mathcal{D}) = \log \mu(\theta) + \sum_{i=1}^{n} \log p(y_i|\mathbf{x}_i, \theta) \quad (7)$$

during training time, and then use this approximate training set posterior $q(\theta)$ in place of the training set when performing inference on the unlabeled test data. This gives us an approximate posterior with log density

$$\log p(\theta|\mathcal{D}, \mathcal{U}) = \log q(\theta) - \frac{\tilde{\alpha}}{m} \sum_{j=1}^{m} H(Y|\tilde{\mathbf{x}}_j, \theta) \quad (8)$$

---

**Algorithm 1** Bayesian Adaptation for Covariate Shift (BACS)

---

**Input**: Ensemble size $k$, Entropy weight $\tilde{\alpha}$, Training Data: $\mathbf{x}_{1:n}, y_{1:n}$, Test Data: $\tilde{\mathbf{x}}_{1:m}$
**Output**: Predictive distributions $p(y|\tilde{\mathbf{x}}_j)$ for each test input $\tilde{\mathbf{x}}_j$
**Training**: For each $i \in (1, \ldots, k)$ compute an approximate density $q_i(\theta) \approx p(\theta|\mathbf{x}_{1:n}, y_{1:n})$
**Adaptation**:
**for all** ensemble members $i \in (1, \ldots, k)$ **do**
    Compute adapted parameters $\hat{\theta}_i = \arg\max_\theta \frac{\tilde{\alpha}}{m} \sum_{j=1}^m -H(Y|\tilde{\mathbf{x}}_j, \theta) + \log q_i(\theta)$
**end for**
For each test input $\tilde{\mathbf{x}}_j$, marginalize over ensemble $p(y|\tilde{\mathbf{x}}_j) = \frac{1}{k} \sum_{i=1}^k p(y|\tilde{\mathbf{x}}_j, \hat{\theta}_i)$.

---

Unlike the full training set, the learned approximate posterior density $q(\theta)$, which can be as simple as a diagonal Gaussian distribution, can be much easier to store and optimize over for test-time adaptation, and the training time procedure would be identical to any approximate Bayesian method that learns a posterior density. In principle, we can now instantiate test-time adaptation by running any approximate Bayesian inference algorithm, such as variational inference or MCMC, to sample $\theta$'s from the density in Equation 8, and average predictions from these samples to compute the marginal probabilities for each desired test label.

**Practical instantiation with ensembles.** As previously mentioned, marginalizing over different models that provide diverse labelings of the test set is crucial to providing uncertainty estimates after adaptation via entropy minimization. We thus propose to use an ensembling approach [Lakshminarayanan et al., 2016] as a practical method to adapt to the test distribution while maintaining diverse labelings. Deep ensembles simply train multiple models from different random initializations, each independently optimizing the target likelihoods, and averages together the models' predicted probabilities at test time. They are able to provide effective approximations to Bayesian marginalization due to their ability to aggregate models across highly distinct modes of the loss landscape [Fort et al., 2019, Wilson and Izmailov, 2020].

Our method, BACS, summarized in Algorithm 1, trains an ensemble of $k$ different models on the training set, each with their approximate posterior $q_i(\theta)$ that captures the local loss landscape around each mode in the ensemble. Then at test time, we independently optimize each of the $k$ models by minimizing Equation 8 (using the corresponding $q_i(\theta)$ for each ensemble member), and then average the predictions across all adapted ensemble members.

## 3 Related Work

**Entropy minimization**. Entropy minimization has been used as a self-supervised objective in many settings, including domain adaptation [Saito et al., 2019, Carlucci et al., 2017], semisupervised learning [Grandvalet and Bengio, 2004, Berthelot et al., 2019, Lee and Lee, 2013], and few-shot learning [Dhillon et al., 2015]. Grandvalet and Bengio [2004] proposed a probabilistic model incorporating entropy minimization for semisupervised learning (without distribution shift), but only use the probabilistic model to motivate entropy minimization as a regularizer for a MAP solution in order improve accuracy, which does not capture any epistemic uncertainty. In contrast, we are concerned with test-time adaptation under distribution shift, which requires introducing a model of separate training-time and test-time input distributions, and with providing reliable epistemic uncertainty estimates, which we obtain via Bayesian marginalization. We also devise an approximate inference scheme to allow for efficient adaptation without access to the training data.

Test time entropy minimization (TENT) [Wang et al., 2020a] uses entropy minimization as the sole objective when adapting to the test data (though without an explicit Bayesian interpretation) and adapts without any further access to the training data, but only aims to improve accuracy and not uncertainty estimation. Similarly to Grandvalet and Bengio [2004], TENT only learns a single model using entropy minimization, whereas we show that explicitly performing Bayesian inference and marginalizing over multiple models is crucial for effective uncertainty estimation. TENT also heuristically proposes to only adapt specific parameters in the networks at test time for stability reasons, while our usage of a learned posterior density to account for the training set allows us to adapt the whole network, improving performance in some settings and eliminating the need for the heuristic design decision.

**Domain Adaptation**. Unsupervised domain adaptation [Ganin et al., 2015, Wang and Deng, 2018] tackles a similar problem of learning classifiers in the presence of distribution shift between our train and test distributions. Most unsupervised domain adaptation works assume access to both the labeled training data as well as unlabeled test data at the same time, while we restrict adaptation to occur without any further access to the training data. One recent line of work, known as source-free domain adaptation [Liang et al., 2020, Kundu et al., 2020, Li et al., 2020] also restricts the adaptation procedure to not have access to the training data together with the unlabeled data from the test distribution. In contrast to these algorithms, we are concerned with using adaptation to improve both uncertainty estimation as well as accuracy, and our algorithm is additionally amenable to an online setting, where prediction and adaptation occur simultaneously without needing to see the entirety of the test inputs.

**Uncertainty estimation under distribution shift (no adaptation)**. Various methods have been proposed for uncertainty estimation with distribution shift that do not incorporate test time adaptation. Ovadia et al. [2019] measured the calibration of various models under various distribution shift without any adaptation, finding that deep ensembles [Lakshminarayanan et al., 2016] and some other Bayesian methods that marginalize over multiple models perform well compared to methods that try to recalibrate predictions using only the source data. Beyond ensembles, other Bayesian techniques [Dusenberry et al., 2020, Maddox et al., 2019] have also demonstrated improved uncertainty estimation under distribution shift compared to standard models. Various other techniques have been found to improving calibration under distribution shift through significant changes to the training procedure, for example utilizing different loss functions [Padhy et al., 2020, Tomani and Buettner, 2020], extensive data-augmentations [Hendrycks et al., 2019], or extra pre-training [Xie et al., 2019b].

**Uncertainty estimation with both train and test distributions**. We now discuss prior work that considers uncertainty estimation assuming access to both train and test data simultaneously. Recent work studying calibration for domain adaptation algorithms [Pampari and Ermon, 2020, Park et al., 2020, Wang et al., 2020b] found that predictions are poorly calibrated in the target domain even if the models were well-calibrated on the labeled source domain. These works all propose methods based on importance weighting between the target domain and the labeled source domain in order to recalibrate target domain predictions using only labels in the source domain. They are not directly applicable in our test-time adaptation setting, since they require estimates of density ratios between target and source distributions, which we cannot obtain without either a generative model of training inputs, or access to the training data during adaptation. Our method also differs in how we approach uncertainty estimation. Instead of using extra labeled data and post-hoc recalibration techniques for classifiers, our method uses Bayesian inference to provide meaningful uncertainty estimates.

For uncertainty estimation in regression problems, Chan et al. [2020] also propose to adapt Bayesian posteriors to unlabeled data by optimizing the predictive variance of Bayesian neural network at an input to serve as a binary classifier of whether the point is in-distribution or not. thus encouraging higher variance for out-of-distribution points. Their method is again not applicable in our test-time setting because they require access to both the train and test data at once.

**Uncertainty estimation with test time adaptation**. Nado et al. [2020] evaluate various techniques for uncertainty estimation in conjunction with adapting batch-norm statistics to the shifted test domain, and again find that deep ensembles provide well-calibrated prediction in addition to improved accuracy. While our method similarly utilizes ensembles and adapts batch norm statistics, and we show that additionally adapting via entropy minimization at test time further improves predictive accuracy without sacrificing calibration.

**Bayesian semi-supervised methods**: Seeger [2000] proposed a probablistic model for incorporating unlabeled data in semi-supervised learning to motivate regularization for the classifier that depends on the input distribution in MAP inference. However, they do not tackle uncertainty estimation and their model does not account for any distribution shift like ours does. Gordon and Hernández-Lobato [2020] perform Bayesian semi-supervised learning combining both generative and discriminative models. In contrast, our method does not need to learn a generative model of the data, and explicitly tackles the problem of distribution shift instead of assuming the labeled and unlabeled data come from the same distribution. Another line of work [Ng et al., 2018, Ma et al., 2019, Walker and Glocker, 2019, Liu et al., 2020] propose Bayesian methods for semisupervised learning specialized graph-structured data.

| | CIFAR10-C | | | | CIFAR100-C | | | |
|---|---|---|---|---|---|---|---|---|
| Method | Acc | NLL | Brier | ECE | Acc | NLL | Brier | ECE |
| Vanilla | 59.90 | 1.892 | 0.6216 | 0.2489 | 35.72 | 4.271 | 0.9797 | 0.3883 |
| BN Adapt | 82.36 | 0.8636 | 0.2909 | 0.1204 | 57.58 | 2.294 | 0.6401 | 0.2377 |
| TENT (1 epoch) | 84.29 | 0.7862 | 0.2629 | 0.1119 | 62.46 | 2.047 | 0.5828 | 0.2280 |
| TENT (5 epoch) | 85.16 | 0.8483 | 0.2603 | 0.1191 | 63.46 | 2.199 | 0.5987 | 0.2592 |
| BACS (MAP) (1 epoch) | 84.82 | 0.7808 | 0.2585 | 0.1119 | 63.05 | 2.090 | 0.5908 | 0.2411 |
| BACS (MAP) (5 epochs) | 85.20 | 0.8075 | 0.2575 | 0.1144 | 63.53 | 2.175 | 0.6009 | 0.2551 |
| Vanilla Ensemble | 61.72 | 1.535 | 0.5431 | 0.1684 | 38.66 | 3.343 | 0.8439 | 0.2462 |
| Ensemble BN Adapt | 85.99 | 0.4722 | 0.2043 | 0.03229 | 64.22 | 1.464 | 0.4793 | **0.0515** |
| Ensemble TENT (1 epoch) | 87.28 | 0.4351 | 0.1867 | **0.02868** | 67.83 | **1.318** | 0.4392 | 0.05909 |
| BACS (ours) (1 epoch) | **87.77** | **0.4260** | **0.1809** | 0.02986 | **68.33** | 1.324 | **0.4360** | 0.06519 |

Table 1: **CIFAR-10/100 Corrupted** results at the highest level of corruption, averaged over all corruption types. With one epoch of adaptation, BACS consistently outperforms all baselines in terms of accuracy, NLL and Brier score, and can further improve with more training. In terms of ECE, all ensembled methods with adaptation performs similarly, and substantially outperforming the non-adaptive or non-ensembled baselines.

## 4 Experiments

In our experiments, we aim to analyze how our test-time adaptation procedure in BACS performs when adapting to various types of distribution shift, in comparison to prior methods, in terms of *both* the accuracy of the adapted model, and its ability to estimate uncertainty and avoid over-confident but incorrect predictions. We evaluate our method and prior techniques across a range of distribution shifts, including corrupted datasets, natural distribution shifts, and domain adaptation settings.

**Architectures and implementation.** For our ImageNet [Deng et al., 2009] experiments, we use the ResNet50v2 [He et al., 2016b] architecture, while for other datasets, we use ResNet26 [He et al., 2016a]. For all methods utilizing ensembles, we use ensembles of 10 models, and report results averaged over the same 10 seeds for the non-ensembled methods. While adapting our networks using the entropy loss, we also allow the batch normalization statistics to adapt to the target distribution. To obtain approximate posteriors for each ensemble member, we use SWAG-D [Maddox et al., 2019], which estimates a Gaussian posterior with diagonal covariance from a trajectory of SGD and requires minimal changes to standard training procedures. During adaptation, we initialize each model from the corresponding posterior mean, corresponding to the solution obtained by Stochastic Weight Averaging [Izmailov et al., 2018]. For methods that optimize on the test distribution, we report results after one epoch of adaptation unless otherwise stated.

**Comparisons.** We compare our method against two state-of-the-art prior methods for test-time adaptation: TENT [Wang et al., 2020a], which simply minimizes entropy on the test data with a single model, without the additional posterior term accounting for the training set that we use in BACS, and ensembles adapted using the batch norm statistics of the shifted test set, as discussed by Nado et al. [2020]. We also compare to deep ensembles [Lakshminarayanan et al., 2016] without any adaptation as a baseline for uncertainty estimation under distribution shift, as well as ensembles of models each adapted using TENT.

**Metrics.** In addition to accuracy, we also evaluate uncertainty estimation using the negative log likelihood (NLL), Brier score [Brier, 1950] and expected calibration error (ECE) [Naeini et al., 2015] metrics. NLL and Brier score are both proper scoring rules [Gneiting and Raftery, 2007], and are minimized if and only if the predicted distribution is identical to the true distribution. ECE measures calibration by binning predictions according to the predicted confidence and averaging the absolute differences between the average confidence and empirical accuracy within each bin.

**Corrupted images.** We first evaluate our method on CIFAR-10-C, CIFAR-100-C, and ImageNet-C [Hendrycks and Dietterich, 2019], where distribution-shifted datasets are generated by applying different image corruptions at different intensities to the test sets of CIFAR10, CIFAR100 [Krizhevsky, 2012], and ImageNet [Deng et al., 2009] respectively. In Table 1, we show comparisons at the most severe level of corruption for CIFAR10-C and CIFAR100-C. With just a single epoch of adaptation at test time, both our method BACS and TENT ensembles substantially outperform other baselines in accuracy, NLL, and Brier score, with BACS improving slightly over TENT ensembles, showing that combining test-time entropy minimization with Bayesian marginalization can lead to strong improvements over either alone. We also note that simply adapting for more epochs with entropy

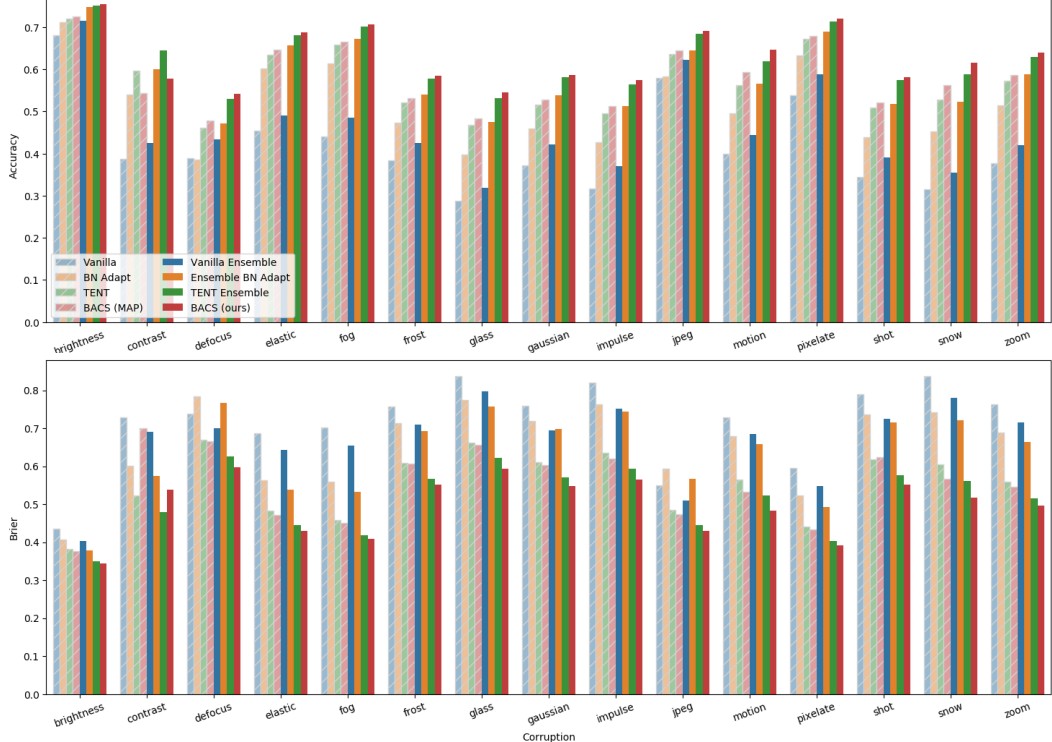

Figure 4: **ImageNet-C Results by Corruption**. For each corruption type, we show the results for each method averaged over all levels of corruption. BACS improves over baselines for every corruption type except for contrast, where BACS has close to 0 accuracy at the most severe level of corruption.

minimization (as seen in TENT (5 epochs)), can further improve accuracy of a single model, but can actually lead to *worse* uncertainty estimates as measured by NLL or ECE as predictions become excessively overconfident. In contrast, combining ensembling with entropy minimization consistently improves accuracy much more than simply adapting a single model for more epochs, while also substantially improving uncertainty estimates.

In Table 2, we show comparisons on ImageNet-C averaged over all corruption levels and corruption types. BACS is able to outperform all prior methods in terms of accuracy, NLL, and Brier score. We note that ensembling and batch norm adaptation actually *worsen calibration*, as measured by ECE, compared to the single model or non-adapted baselines respectively, despite each technique providing substantial improvements

| Method | Acc | NLL | Brier | ECE |
|---|---|---|---|---|
| Vanilla | 41.79 | 3.127 | 0.7152 | 0.06439 |
| BN Adapt | 51.54 | 2.676 | 0.6564 | 0.1709 |
| TENT | 57.06 | 2.035 | 0.5536 | **0.0342** |
| BACS (MAP) | 58.06 | 2.036 | 0.5528 | 0.04270 |
| Vanilla ensemble | 46.03 | 2.788 | 0.6670 | 0.07256 |
| Ensemble BN Adapt | 58.30 | 2.428 | 0.6333 | 0.2594 |
| Ensemble TENT | 62.40 | 1.788 | 0.5131 | 0.1010 |
| BACS (ours) | **63.04** | **1.726** | **0.4962** | 0.08308 |

Table 2: **ImageNet-C** results averaged over all corruption types and levels. BACS outperforms all baselines in accuracy, NLL, and Brier scores.

on all other metrics. We see that an ablation of our method that only uses a single model, BACS (MAP), outperforms the other non-ensemble methods in accuracy, NLL, and Brier score.

We also show results for accuracy and Brier score at each level of corruption in Figure 4. BACS (ours) consistently outperforms baselines at each level of corruption (with one exception being accuracy at the highest level of corruption, where a single corruption with much lower accuracy drags down the mean to be slightly below to that of Ensemble TENT).

**ImageNet-R.** In Table 3, we further evaluate robustness using ImageNet-R [Hendrycks et al., 2021], which consists of images that are abstract renditions of 200 of the ImageNet classes. Similar to our Imagenet-C results, we find that BACS performs the best across accuracy, NLL, and Brier score, while being slightly outperformed in ECE by vanilla ensembles.

**Impact of posterior term.** In this section, we discuss the effects of using the training set posterior density in our objective (Equation 8) when adapting at test time. Intuitively, the training posterior density ensures that our adapted classifiers, while minimizing entropy on the target distribution, remain constrained to still perform well on the training set and stay near the initial solution found during training.

| Method | Acc | NLL | Brier | ECE |
|---|---|---|---|---|
| Vanilla | 36.40 | 3.288 | 0.7602 | 0.05667 |
| BN Adapt | 38.05 | 3.236 | 0.7646 | 0.09744 |
| TENT | 41.13 | 2.967 | 0.7167 | 0.02666 |
| BACS (MAP) | 43.55 | 2.909 | 0.7183 | 0.1074 |
| Vanilla ensemble | 40.38 | 3.011 | 0.7180 | **0.02339** |
| Ensemble BN Adapt | 42.82 | 3.019 | 0.7408 | 0.1696 |
| Ensemble TENT | 45.75 | 2.726 | 0.6815 | 0.1025 |
| BACS (ours) | **47.31** | **2.565** | **0.6625** | 0.04270 |

Table 3: **ImageNet-R** results. BACS outperforms all baselines in accuracy, NLL, and Brier scores.

We empirically find that the posterior term can be important for preventing entropy minimization from finding degenerate solutions: adapting the whole network on several ImageNet-C corruptions without the posterior term can lead to poor models that achieve close to 0 accuracy on many corruptions.

While TENT, which adapts via entropy minimization without any term accounting for the training data, uses an ad-hoc solution that restricts adaptation to only the learnable scale and shift parameters in the batch norm layers, our use of the training set posterior density is motivated directly from our proposed probabilistic model and does not require any heuristic choices over which parameters should or should not be adapted. In our ImageNet experiments (Tables 2 and 3), we see that an ablation of our method without ensembles, corresponding to the MAP solution in our proposed model, outperforms TENT. The difference between our ablation BACS (MAP) and TENT is precisely that BACS (MAP) adapts the whole network, but with an additional regularization term, while TENT adapts only a small part of the network without any regularizer.

However, we find the approximate posterior density can also be limiting in certain situations, which we can observe in experiments transferring from SVHN to MNIST (Appendix B.4). Here, there is a large discrepancy between the training and test domains, and we find that adaptation with the posterior is unable to adjust the parameters enough and underperforms compared to an ablation of our method that simply removes the posterior term while still adapting the whole network.

# 5 Discussion

We presented Bayesian Adaptation for Covariate Shift (BACS), a Bayesian approach for utilizing test-time adaptation to obtain both improved accuracy and well-calibrated uncertainty estimates when faced with distribution shift. We have shown that adapting via entropy minimization without Bayesian marginalization can lead to overconfident uncertainty estimates, while our principled usage of an approximate training posterior during adaptation can outperform previous heuristic methods. These observation support our hypothesis that framing entropy minimization within a well-defined Bayesian model can lead to significantly more effective test-time adaptation techniques. Our method is straightforward to implement, requires minimal changes to standard training procedures, and improves both accuracy and uncertainty estimation for a variety of distribution shifts in classification.

**Limitations and future work.** One limitation of our approach is that it requires effective techniques for estimating the parameter posterior from the training set. While the study of such methods, and Bayesian neural networks more broadly, is an active area of research, it remains a significant practical challenge. It is likely that the simple Gaussian posteriors we employ provide a poor estimation of the true posterior and can overly constrain the network during adaptation. Therefore, a relevant direction for future work is to integrate BACS with more sophisticated Bayesian neural network methods.

Another promising direction for future work is to explore other objectives that have had success in semi-supervised learning settings, such as consistency based losses [Sajjadi et al., 2016, Miyato et al., 2017, Xie et al., 2019a, Sohn et al., 2020] or information maximization [Gomes et al., 2010], which can be straightforwardly incorporated into our method as suitable priors on the relationship between the data distribution and classifier. More broadly, we hope that our work will spur further research into test-time adaptation techniques based on well-defined Bayesian models that describe how unlabeled test data should inform our posterior estimates of the model parameters, even in the presence of distributional shift.

## Acknowledgements

We thank the anonymous reviewers for their extremely valuable feedback and discussions, which have greatly improved our paper. This research was supported by the DARPA Assured Autonomy program and DARPA LwLL, with compute support from Google Cloud and the Tensorflow Research Cloud (TFRC) program.

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
