# OpenReview forum: "Bayesian Adaptation for Covariate Shift"
_NeurIPS.cc/2021/Conference — NeurIPS 2021 Poster_

### Official Review · Reviewer_6Ge4 · 2021-07-04

**Rating:** 6
**Confidence:** 3

**Summary:**

This paper presents a Bayesian method for handling a covariate shift in a fully test-time adaptation setting. The authors build on Seeger (2000) to incorporate the unlabeled test examples in their model with an entropy-based empirical prior. They propose a graphical model having three latent variables of interest, the first controls the conditional distribution of the target given the data ($\theta$), and two variables that govern the input distributions, one for the training data ($\phi$) and the other for the test data ($\tilde{\phi}$). Posterior inference is done only for $\theta$ while using a point estimate for $\phi, \tilde{\phi}$ with empirical Bayes at the observed data points. To obtain the posterior, first, an approximation of it is obtained using the training data, and second, the approximation is corrected with the test data using an entropy-based term. The authors compared their method against several baselines on corrupted datasets and domain adaptation benchmarks.

**Limitations And Societal Impact:**

The authors addressed the potential limitations and negative societal impact of their work.

**Main Review:**

The paper presents the following merits:
- Using the Bayesian formalization proposed in the paper makes sense and seems to be original in that context.
- The derivation in the paper is elegant and results in an intuitive posterior density.
- The resulting adaptation strategy is simple. It is great that training can be done efficiently with standard supervised techniques and that the test-time adaptation is not computationally intensive (when ignoring the costs imposed by using deep ensembles).
- The method shows the best results (in both accuracy and uncertainty quantification) in almost all cases against the compared baselines.
- Most experimental details required for reproducing the results were given. Code was submitted as well, which is noteworthy.

Concerns/limitations:
- The main concern I have with the paper is that it ignores relevant domain adaptation (DA) studies in the literature review and the experimental section. More specifically, it ignores papers that fall under the umbrella of source-free DA/DA without source data (e.g., [1, 2, 3]). In this setup as well the classifier is initially presented with examples from one domain and no longer has access to them after learning the model and when encountered with examples from a new domain. I think that the authors should have addressed this line of work and evaluate their method against top baselines in this field, at least in the offline setting.
- The method suggested in this paper operates under the assumption that the training data is not available during adaptation. Although I acknowledge the importance of this setting, often in real-life scenarios it is possible to maintain and make use of at least some of the training examples. I think that a discussion on how to use BACS in this setting is missing.
- I appreciate the diversity of experiments and the datasets used; however, for the DA part more challenging datasets could have been chosen (for example, datasets from Office-Home [4] or ViSDA-C [5]).
- Some experimental details are not clear from the text. (1) How did you chose alpha (or conversely beta) and the number of epochs for adaptation in each experiment. Did you have a validation set? If so, did you perform a grid search?  (2) More details about the corrupted datasets than just specifying the references will be great. (3) Did you need to tune hyper-parameters for baseline methods as well? if so, how was it done?

Minor comments:
- Some references should be altered from arXiv to the publication venue (e.g., [6, 7]).
- According to NeurIPS guidelines, tables should not contain vertical rules.
- It is not clear from the text what is the vanilla baseline.
- Some typing mistakes: index is over i in the last line of Algorithm 1, "is able improves" (L319), "We also include show tables" (L675).

Overall it is a nice paper with solid results (compared to the baselines included). I think that the Bayesian treatment is justified. There are some issues with the paper, mainly in the experimental section, that should be addressed.

[1] Li, R., Jiao, Q., Cao, W., Wong, H. S., & Wu, S. (2020). Model adaptation: Unsupervised domain adaptation without source data. In Proceedings of the IEEE/CVF Conference on Computer Vision and Pattern Recognition (pp. 9641-9650).
[2] Kundu, J. N., Venkat, N., & Babu, R. V. (2020). Universal source-free domain adaptation. In Proceedings of the IEEE/CVF Conference on Computer Vision and Pattern Recognition (pp. 4544-4553).
[3] Liang, J., Hu, D., & Feng, J. (2020, November). Do we really need to access the source data? source hypothesis transfer for unsupervised domain adaptation. In International Conference on Machine Learning (pp. 6028-6039). PMLR.
[4] Venkateswara, H., Eusebio, J., Chakraborty, S., & Panchanathan, S. (2017). Deep hashing network for unsupervised domain adaptation. In Proceedings of the IEEE conference on computer vision and pattern recognition (pp. 5018-5027).
[5] Peng, X., Usman, B., Kaushik, N., Hoffman, J., Wang, D., & Saenko, K. (2017). Visda: The visual domain adaptation challenge. arXiv preprint arXiv:1710.06924.
[6] Sun, Y., Wang, X., Liu, Z., Miller, J., Efros, A., & Hardt, M. (2020, November). Test-time training with self-supervision for generalization under distribution shifts. In International Conference on Machine Learning (pp. 9229-9248). PMLR.
[7] Xie, Q., Dai, Z., Hovy, E., Luong, T., & Le, Q. (2020). Unsupervised Data Augmentation for Consistency Training. Advances in Neural Information Processing Systems, 33.

**Time Spent Reviewing:**

10

---

> ### Author Response · Authors · 2021-08-10
> **Response to Reviewer 6Ge4**
>
> **Experimental Details**: Please see the shared reply to all reviewers for details on hyperparameter selection for each experiment, as well as additional results with more fine grained hyperparameter tuning on BACS as well as the baseline TENT. We will also include detailed descriptions of the corruptions evaluated in the appendix.
>
> **Source-free domain adaptation**: We will include discussion of domain adaptation and especially source-free domain adaptation in related work. We emphasize that our method, similar to TENT and batch norm adaptation, is primarily aimed at improving robustness rather than at classic domain adaptation settings, and is very easy to use with minimal changes to training procedures.
>
> In contrast, the source-free domain adaptation algorithms the reviewer can introduce substantial changes to the training procedures compared to typical supervised learning procedures.. For example, the methods proposed in Li et al [1] and Kundu et al [2] require training generative models. SHOT [3] is comparably simpler, though still requires a pseudolabeling procedure in addition to an information maximization loss for source free adaptation. We will add comparisons to SHOT in our offline evaluations, though we also emphasize that our method, similar to TENT, is amenable to an online evaluation setting, while the source-free domain adaptation algorithms like SHOT are not.
>
> **Maintaining some training examples**: We agree with the reviewer that the setting where we are able to maintain access to some amount of training data during adaptation is an interesting and useful problem to tackle. One way to use BACS in this setting is to include both the posterior (with reduced weight) as well as the limited labeled training data to constrain optimization, which may allow for a more accurate approximation to the training set and improve performance. We will add a brief discussion of this setting in our paper, but we leave a thorough evaluation of this novel problem setting to future work.
>
> [1] Li, R., Jiao, Q., Cao, W., Wong, H. S., & Wu, S. (2020). Model adaptation: Unsupervised domain adaptation without source data. In Proceedings of the IEEE/CVF Conference on Computer Vision and Pattern Recognition (pp. 9641-9650).
>
> [2] Kundu, J. N., Venkat, N., & Babu, R. V. (2020). Universal source-free domain adaptation. In Proceedings of the IEEE/CVF Conference on Computer Vision and Pattern Recognition (pp. 4544-4553).
>
> [3] Liang, J., Hu, D., & Feng, J. (2020, November). Do we really need to access the source data? source hypothesis transfer for unsupervised domain adaptation. In International Conference on Machine Learning (pp. 6028-6039). PMLR.

---

> > ### Comment · Reviewer_6Ge4 · 2021-08-19
> > **Response to Author's Rebuttal**
> >
> > Thank you for the response. I still believe that the compared baselines are not sufficient and that the DA experiments should have been conducted on more challenging datasets. Nevertheless, as stated in my original review, this paper presents other merits and should be accepted for them. Therefore, I will keep my original score intact.

---

### Official Review · Reviewer_RDBC · 2021-07-15

**Rating:** 6
**Confidence:** 4

**Summary:**

This paper proposes a Bayesian inference framework for test-time adaptation under covariate shift setting.  The setting discussed in the paper is, we have a model trained in a fully supervised way, and observe unlabelled test data (without labels) that might have come from a distribution different from the training data distribution. The goal is to (quickly) adapt the model for the test data without access to the test labels. The proposed method also tries to properly model the epistemic uncertainty of the model to be adapted for the test data, and for this purpose, the problem is formulated as a Bayesian hierarchical model with the data generating distribution and the parameter generating distributions are coupled. Here, an important difference from the previous approaches is that for test data we have a different data generating distribution (conditioning on a different parameter for data generating distribution). The dependence between the model parameter distribution and data generating distribution is expressed in terms of energy-based density whose energies are written as predictive entropies.  The proposed approach entitled Bayesian Adaptation for Covariate Shift (BACS) is demonstrated to be effective in various covariate shift benchmark datasets.

**Limitations And Societal Impact:**

The paper properly discusses the limitation in section 5. I agree with the authors that the proposed approach does not have a negative societal impact.

**Main Review:**

I enjoyed reading the paper, it is clearly written and easy to follow. The related work section is extensive and faithfully covers the literature.

I like the way the authors presented the covariate shift setting as a Bayesian model with coupled dependence between data generating distribution and the parameter distribution (more specifically, the distribution $p(\theta|\phi, \tilde\phi)$.  It is a classic trick to model such distribution as an energy-based model with entropy energy function (as authors stated in the paper, for example, Grandvalet and Bengio 2014), but the modification with test-time parameter $\tilde\phi$ is an interesting extension. The optimization objective derived as a result can be interpreted as a probabilistic version of TENT (Wang et al 2020a) without the regularization term originated from the model posterior obtained from the training set.  The resulting algorithm seems to be easy to implement.

The experimental results are extensive and convincing. BACS outperformed the baselines for most of the experiments done in terms of both prediction accuracy and uncertainty estimate. The appendix provides further results, especially the results under various corruption levels.

Questions & comments

1. I guess the hyperparameter $\tilde\alpha$ (or $\beta = 1/\tilde\alpha)$ plays an important role in controlling the behaviour of the algorithm. However, judging from the main text and the appendix, the paper does not provide a principled way of choosing it. How did you select the value $\beta$? It would be hard to apply conventional tuning methods such as cross-validation or Bayesian optimization under test-time adaptation. Also, how sensitive the algorithm is to the choice of $\beta$?

2. In SVHN to MNIST experiments, BACS slightly underperformed against TENT in terms of prediction accuracy. This is maybe because of the amount of shift between SVHN and MNIST; since MNIST is quite different from SVHN, it requires more adaptation, so the regularization with the model posterior basically requires the adapted parameters to be close to the original model posterior parameters. The authors additionally showed that BACS without the posterior regularization (BACS-posterior) outperforms TENT, and this also implies the sensitivity of BACS to the choice of the parameter $\beta$. Have you tested with different values of $\beta$ for this setting? Also, I guess BACS-posterior can serve as a decent baseline to compare. How does it compare against BACS for the other experiments?

3. Often, we don't even know whether given test data is indeed from a shifted distribution. I think it would be interesting to apply BACS to the test data coming from the distribution identical to the training data distribution and see how it affects the performance.

4. I think the role of the posterior regularization (by the term $\log q_i(\theta)$) is reminiscent of the regularization techniques used in continual learning, where the goal is to prevent parameters deviate too much from the original one and thus result in catastrophic forgetting.  It would thus be interesting to consider the regularization terms used in continual learning literature for the test time adaptation problem.




**Time Spent Reviewing:**

6 hours

---

> ### Author Response · Authors · 2021-08-10
> **Response to Reviewer RDBC**
>
> **In-distribution performance**: We evaluate methods on the in-distribution test sets of CIFAR10/100, and find that all ensemble methods perform similarly to one another, while all the non-ensembled methods also perform similarly. We do find that adapting via entropy minimization can slightly hurt in-distribution performance, though we believe this to be outweighed by the substantial improvements in robustness under distribution shift.
>
> | CIFAR10          | Acc   | NLL    | Brier   | ECE      |
> |------------------|-------|--------|---------|----------|
> | Vanilla          | 95.50 | 0.1715 | 0.07252 | 0.02549  |
> | Vanilla Ensemble | 96.07 | **0.122**  | **0.05835** | 0.009644 |
> | BN Adapt         | 95.49 | 0.1767 | 0.07303 | 0.02588  |
> | BN Ensemble      | **96.09** | 0.1244 | **0.05834** | **0.009433** |
> | TENT             | 95.48 | 0.1788 | 0.07662 | 0.03180  |
> | BACS (ours)      | 95.98 | 0.1340 | 0.06154 | 0.01091  |
>
> | CIFAR100         | Acc   | NLL    | Brier  | ECE     |
> |------------------|-------|--------|--------|---------|
> | Vanilla          | 77.88 | 1.023  | 0.3389 | 0.1198  |
> | Vanilla Ensemble | 80.34 | **0.7182** | **0.2711** | **0.04254** |
> | BN Adapt         | 77.88 | 1.071  | 0.3437 | 0.1266  |
> | BN Ensemble      | **80.46** | 0.7312 | 0.2720 | 0.04438 |
> | TENT             | 77.86 | 1.083  | 0.3458 | 0.1322  |
> | BACS (ours)      | 80.32 | 0.7428 | 0.2754 | 0.05051 |
>
> **Hyperparameter selection**: See the reply addressed to all reviewers for a discussion of hyperparameter selection and improved results upon more fine-grained tuning of hyperparameters using the ImageNet-C validation corruptions.
>
> **BACS - posterior performance**: We found BACS - posterior was unable to perform well on ImageNet-C. On certain corruptions in the validation corruption set, the adaptation procedure (with no constraint or limitation on what parameters were adapted) quickly dropped to near 0 accuracy.
>
> On SVHN-> MNIST, we do find that reducing $\beta$ (putting more weight on the test entropy) does improve the performance of BACS. Setting $\beta =0.00001$ (10x lower than in our reported results) gives an accuracy of 86.4 after 1 epoch and 91.3 after 10 epochs, which improves over the accuracies we reported for BACS with $\beta=0.0001$ (85.04 and 86.56 respectively), but still worse than removing the posterior term entirely, which results in accuracies of 89.30 and 94.07 respectively.

---

> > ### Comment · Reviewer_RDBC · 2021-08-20
> > **Thanks for the response**
> >
> > I appreciate the authors' effort to address my concerns, and also the response to the other reviewers in general. My main concern on hyperparameter selection and uncertainty evaluation metrics has been resolved, so I'd like to keep my score intact.

---

### Official Review · Reviewer_EWK6 · 2021-07-16

**Rating:** 7
**Confidence:** 5

**Summary:**

The paper extends test-time entropy minimization through a novel Bayesian criterion for adaptation. It allows to improve performance during adaptation without degrading the model calibration, a shortcoming of standard entropy minimization. The model is evaluated on CIFAR-10C/-100C/ImageNet-C and compared to batch norm adaptation and test-time entropy minimization.

**Limitations And Societal Impact:**

The paper discusses limitations to a sufficient extend in the conclusion section.

**Main Review:**

The method is overall interesting and I think with a bit of improvements in the writing, the experiments and analysis, this could become a nice contribution. As of now, the results are inconclusive and ablations are lacking given that the loss function introduces a few new hyperparameters.

Major:
* The method section needs work. The key idea of the method is currently hidden behind notation and theoretical motivations which are not experimentally verified and analysed. I realize this is matter of taste, but I am in favor of clearly stating out the learning algorithm, highlighting the differences to TENT-only training (ensembles, Bayesian log q(\theta) update, and then putting the theoretical justification afterwards. The actual practical difference in the loss function to TENT seems minimal (yet elegant!) and this should be highlighted, I think it is actually a selling point of the method.
* For the key table (Table 2) on large scale data, the method does not seem to fully work (?). The accuracy is only minimally better than TENT (and one issue of the TENT paper is that it does not contain a hyperparameter sweep for ImageNet scale experiments, so the true potential performance will likely be a bit higher than 58%) and the ECE is actually worse than the vanilla network. How would the model selection take place in practice? The model needs to be selected without access to any of the numbers present in Table 2. Also, I would find it important to show mean-corruption error (mCE, see comment further below) apart from accuracy.
* Why does Table 1 only show the results at the highest corruption level? Could you provide the full picture by providing these numbers on all corruptions?
* l. 598: It is unclear how the hyperparameters are chosen, especially since they differ from those reported in TENT. Also the new hyperparameter beta is simply selected as 0.001 for small scale experiments and 0.0001 for ImageNet-C, and the early stopping (after 1 vs. 5 vs. 10 epochs) is also arbitrarily selected. I assume good intent and that the authors simply did not test any other setting. If they did, they should include this information and state in case they picked the hyperparameter based on the test performance (which I would accept if it is stated in the main paper). An even better solution would be an ablation on the development corruptions in CIFAR-C/ImageNet-C (this would be 4 corruptions x 5 severities), and selecting beta based on the best accuracy in this dataset. An ablation across beta on said development set would be interesting as I think it will have an impact on the final performance.
* On ImageNet-C, the *mean corruption error* is a very common metric and indeed proposed as the metric in the official paper. The authors should report this metric in the supplement and/or provide results that make it possible for others to compute it (i.e., at the very minimum, per-corruption accuracies/errors averaged across severities, even better would be a table with all corruptions and severities, or simply computing the mCE). It might also be that mCE will change the model ranking in the ImageNet-C table.
* I think the comparison to TENT, at the very least its performance, is not fair. The author’s method uses ensembling over 10 models, while only a single TENT model is considered. It is well-known that ensembling improves accuracy. The authors should include results for adapting 10 models with TENT as a baseline. Even if that removes the improvement in terms of accuracy, I would expect that the calibration result will hold true (?)
* The ECE score is sensitive to the number of bins. Could the authors run the following re-analysis of the results: Test bin sizes [5, 10, 20, 30, 50, …] and report whether this changes the scores on ImageNet-C in a non-trivial way? What was the rationale for picking 20 bins?
* A ResNet-26 has 17.96 million parameters. In single precision float32, this amounts to 70Mb in weights and 35Mb in half-precision float16. SVHN train is 174Mb large, CIFAR 162Mb. In the light of this, I find it hard to disregard all previous work in domain adaptation on the basis of doing “test-time” adaptation. When the model is only barely smaller than the datasets.
  * CIFAR10 and SVHN->MNIST results: Using BN adaptation and TENT on these datasets are outdated baselines. Both algorithms were proposed on robustness settings, not primarily on these classical domain adaptation scenarios. Methods like self-ensembling (French 2017), DIRT-T (Shu 2018) and follow up methods could be potential pointers.

Suggestions:
* I would propose to make a plot that better shows the trade-off between calibration and accuracy.
* On one of the smaller scale dataset, could you perform the following experiment: Take TENT, compute mCE on the development corruptions, and tune based on ECE and accuracy (jointly using a suitable criterion). What is the upper bound performance of your baseline?
* Consider including results on ImageNet-R for an instance of more "natural" distribution shift.
* Ablations should be added for all newly introduced parameters, ideally on ImageNet scale (e.g. by running on the development corruptions in ImageNet-C)

Minor:
* l. 100 It is not clear why $\theta$ would influence the data distribution in Figure 3.
* l. 179 How are the predictions aggregated?
* How do BACS (ours), BACS (MAP) and BACS - posterior compare?
* I would recommend to replace supplementary figure 4 with violin or scatter plots instead of box plots, to better show the differences between the (very different) various domains in ImageNet-C.
* The graphical models are misleading. It should be stated that the graphical models depict the predictive distribution, not the dataset distribution (since $\theta$ -> $y$).
* Table typesetting could be improved, e.g. by removing vertical lines in all tables.

### Update of original review score based on author responses

The paper considerably improved during the rebuttal period, most notably for me is
1. the commitment to get results on the "real"/frozen ImageNet-C test set for better comparability to other data for the camera ready version,
2. the inclusion of strong ImageNet-R results,
3. the addition of ensembling baselines for TENT and improvement of tables/figures in this regard,
4. the commitment to situate the small scale domain adaptation results better in the existing literature (and possible tune down this section of the paper to make room for a more extensive discussion of the ImageNet-scale results).

In the light of the new results provided during the rebuttal phase, the discussion with the other reviewers, as well as the promise to add revised ImageNet-C results for the camera-ready, I decided to increase my score from (5) to (7) and recommend to accept the paper; I also increased the confidence score from (4) to (5). I thank the authors for the great discussion.

**Time Spent Reviewing:**

3

---

> ### Author Response · Authors · 2021-08-10
> **Response to Reviewer EWK6**
>
> Thank you for your comments and suggestions. In this response, as well as the general response to all reviewers, we add almost all of the requested experiments, with the exception of the additional corruption levels on CIFAR10/100-C, which we are running but which did not complete in time for the initial response. We hope that these additional evaluations address the most major concerns in your review, and we welcome suggestions for any other experiments.
>
> || Reviewer EWK6: “For the key table (Table 2) on large scale data, the method does not seem to fully work (?). The accuracy is only minimally better than TENT”
>
> **Response to ImageNet-C performance**: In table 2, we report accuracies of 65.44 for BACS (ours) compared to 58.05 for TENT, and 59.98 for an ablation of our method BACS (MAP) which does not use ensembling. Measuring with mCE instead (see table below), BACS (ours) achieves 43.88% while TENT achieves 53.15% mCE. We believe the improvements in accuracy and mCE for BACS (ours) over TENT are substantial.
>
> **Hyperparameters and baseline tuning**: See the reply addressed to all reviewers for a discussion of hyperparameter selection and improved results upon more fine-grained tuning of hyperparameters (for both BACS and TENT) on the ImageNet-C validation corruptions.
>
> **Comparison with TENT-Ensembles**:  As requested, we include comparisons on ImageNet-C between BACS (ours) and TENT (ensemble), which also ensembles over 10 models. We see that BACS (ours) outperforms TENT (ensemble) in all metrics. (See the shared reply to all reviewers for another comparison after tuning hyperparameters more finely on the validation corruptions).
>
> |                       | Acc   | NLL   | Brier  | ECE    |
> |-----------------------|-------|-------|--------|--------|
> | TENT (ensemble) (old) | 63.77 | 1.857 | 0.5218 | 0.1655 |
> | BACS (ours) (old)     | 65.44 | 1.702 | 0.4542 | 0.1479 |
>
> **Extra ECE calculations**:
> We rerun analysis of ImageNet-C results using different numbers of bins for measuring ECE. Overall, results are very similar across different numbers of bin, and do not affect the relative ranking of methods. We chose 20 bins following the SWAG paper [1].
>
> |                  | 10      | 20 (in main paper) | 30      | 50      | 100     |
> |------------------|---------|--------------------|---------|---------|---------|
> | Vanilla          | 0.06122 | 0.06144            | 0.06167 | 0.06202 | 0.06289 |
> | Vanilla Ensemble | 0.07869 | 0.07878            | 0.07887 | 0.07901 | 0.07940 |
> | TENT             | 0.08123 | 0.08137            | 0.08140 | 0.08152 | 0.08185 |
> | TENT (ensemble)  | 0.1655  | 0.1655             | 0.1655  | 0.1655  | 0.1655  |
> | BACS (MAP)       | 0.06647 | 0.06642            | 0.06671 | 0.06687 | 0.06730 |
> | BACS (ours)      | 0.1479  | 0.1479             | 0.1480  | 0.1480  | 0.1481  |
> | BN Adapt         | 0.1606  | 0.1606             | 0.1606  | 0.1606  | 0.1606  |
> | Ensemble BN      | 0.2438  | 0.2438             | 0.2438  | 0.2438  | 0.2438  |
>
> **mCE results**: We compute mCE for all methods below. We see that the ranking of methods does not change compared to using average accuracy. We also include the mCE results from TENT (ensemble) and BACS (ours) after more fine grained tuning (see reply to all reviewers for details). We will add these mCE results in the appendix as suggested by the reviewer.
>
> | Method                  | mCE (lower is better) |
> |-------------------------|-----------------------|
> | Vanilla                 | 68.43%                |
> | Vanilla Ensemble        | 63.02%                |
> | BN Adapt                | 55.79%                |
> | Ensemble BN Adapt       | 47.95%                |
> | TENT                    | 53.15%                |
> | TENT (ensemble)         | 45.96%                |
> | BACS (MAP)              | 50.78%                |
> | BACS (ours)             | 43.88%                |
> | TENT (ensemble) (tuned) | 43.44%                |
> | BACs (ours) (tuned)     | **42.41%**                |
>
> **CIFAR10/100-C results across extra corruptions**: We reported results only on the highest corruption level to match the results presented in the TENT paper. We will also add CIFAR10/100-C results across all corruptions and grouped by corruption level in the appendix, Unfortunately, we did not have time to run them during the initial review period due to the additional experimental results and ablations we’ve included.
>
> **Method Section Writing**: As suggested, we will add a paragraph in the methods section highlighting the key practical differences between BACS and TENT.
>
> **Domain Adaptation and Test Time Adaptation**:
> We agree with the reviewer that our small scale experiments are not particularly realistic scenarios for showcasing the advantages of test-time adaptation over methods that use the source data. We do emphasize that our large scale experiments on ImageNet-C are a setting where test-time adaptation setting is much more relevant. Additionally, we also evaluate our method in an online setting in appendix B.1, while domain adaptation algorithms typically are not amenable to the online setting.
>
> Regarding baselines on SVHN->MNIST, we emphasize that our method, similar to BN adaptation and TENT, is also primarily aimed at improving robustness, and not specifically for classic domain adaptation settings. We will clarify that the methods evaluated do not achieve state-of-the-art performance on the SVHN->MNIST task, as well as add discussion of domain adaptation in the related work. We also note that the TENT paper does compare against some classic domain algorithms that use both the training and test data, and outperforms them on CIFAR10/100-C.
>
> [1] Maddox, W., Garipov, T., Izmailov, P., Vetrov, D., Wilson, A. A Simple Baseline for Bayesian Uncertainty in Deep Learning. In Proceedings of Advances in Neural Information Processing and Systems Conference 32 (NeurIPS 2019)  https://arxiv.org/abs/1902.02476

---

> > ### Comment · Reviewer_EWK6 · 2021-08-11
> > **Short comment on unsupervised DA**
> >
> > Dear authors,
> >
> > thanks a lot for the response and addressing my key concerns, the updated results look great. I will write a more thorough response in the coming days, but wanted to already add a few statements regarding your last paragraph on unsupervised DA.
> >
> > > We also note that the TENT paper does compare against some classic domain algorithms that use both the training and test data, and outperforms them on CIFAR10/100-C.
> >
> > While this is true, it should be noted that the methods referenced for domain adaptation in the TENT paper are outdated by a few years---in particular, successful DA approaches based on pseudo-labeling (which are assentially a combination of entropy minimization and CE minimization on the source domain and should perform strictly better) are not listed. For example, your best method improves SVHN -> MNIST transfer from 79.52% -> 94.07%, which to the best of my knowledge is worse than the state-of-the art in DA in terms of performance.
> >
> > A slightly outdated algorithm to compare to is [French et al (ICLR 2018)](https://arxiv.org/pdf/1706.05208.pdf) who report 68.65% -> 99.18% performance (Table 1). CIFAR10 to STL10 is not directly comparable since the baseline used in BACS is stronger (BACS achieves 84.03% -> 85.47% while the referenced paper achieves  75.2% -> 80.08%).
> >
> > Even though the TENT paper did not address this, I think it is worth pointing out that BACS does not outperform the state of the art in domain adaptation in terms of accuracy.

---

> > > ### Author Response · Authors · 2021-08-13
> > > **Response on unsupervised DA**
> > >
> > > We thank the reviewer again for the helpful pointers. We will acknowledge that BACS and other methods currently evaluated do not achieve state-of-art-performance in accuracy in these classic domain adaptation settings and add discussion of domain adaptation algorithms in the related work. Please let us know if this addresses your concerns about unsupervised domain adaptation, or if you have any suggestions on other things to include.

---

> > > ### Author Response · Authors · 2021-08-20
> > > **Followup**
> > >
> > > We'd like to thank the reviewer again for the many helpful comments and suggestions. We would greatly appreciate if you could let us know whether you have any remaining concerns after the additional experimental results we've included.

---

> > ### Comment · Reviewer_EWK6 · 2021-08-30
> > **Question about your mCE computation**
> >
> > Dear authors,
> >
> > I have a small question for interpreting your numbers, and the following is regarding the table below "mCE results".
> >
> > You report a score of 68.43% mCE for the vanilla (ResNet50?) network and 55.79% mCE after batch norm adaptation. Could you confirm that these numbers are correct?
> >
> > I acknowledge some variation in the literature due to the choice of the baseline model, [Nado et al. (cf. §5.2)](https://arxiv.org/pdf/2006.10963.pdf) report 60.28% mCE after batch norm adaptation while [Schneider et al. (Neurips 2020, cf. Table 1)](https://arxiv.org/pdf/2006.16971.pdf) report 62.2% mCE for a 76.7% mCE baseline model.
> >
> > Still, 55.79% mCE after batch norm adaptation sounds more like a model trained with additional means of augmentation---but I then the 53.15% mCE you report for TENT is close to the value for an actual "vanilla" ResNet50, the value reported in the TENT paper is 53.5%.
> >
> > Could you clarify why your baseline model is this good and why batch norm adaptation works this well? Specifically:
> > - What kind of data pre-processing / data augmentation is used for training your baseline?
> > - In table "mCE results" above, are all models based on the exact same baseline checkpoints?
> > - Can you verify that mCE above is reported on the 15 test corruptions only, and that you first average across severities, then scale by AlexNet scores, and then average the results as in [Hendrycks & Dietterich (ICLR 2019)](https://arxiv.org/abs/1903.12261)?
> >
> > Thanks for clarifying.

---

> > > ### Author Response · Authors · 2021-08-30
> > > **mCE clarifications**
> > >
> > > As mentioned in the supplementary material, model training for the ImageNet experiments was done using an example training script from DeepMind’s Haiku library. Model training details and the source code can be found in https://github.com/deepmind/dm-haiku/tree/main/examples/imagenet.
> > >
> > > We confirm that the mCE values given for the vanilla model and batchnorm adaptation are correct. We also note that the improved baseline performance is also reflected in other metrics, such as the (unnormalized) accuracy rates averaged across corruptions. One possible reason for the stronger baseline ResNet50 performance is that the example script by default uses ResNetv2 (also known as preactivation ResNet, details of which can be found in https://github.com/deepmind/dm-haiku/blob/main/haiku/_src/nets/resnet.py), which we will clarify in our paper.
> > >
> > > Another small difference in training is that after the initial training loop on ImageNet for 90 epochs, we ran another 10 epochs of training to collect iterates for the SWAG posterior, and initialize all models from the mean of the SWAG posterior (corresponding to a solution found by stochastic weight averaging (SWA)). However, we did verify that both the SWA solution and the solution found after the initial 90 epochs of SGD both had very similar performance without adaptation and with BN adaptation, so we do not believe the usage of SWA to be the reason for the discrepancy in baseline results.
> > >
> > > With regards to the bulleted specific clarifications:
> > >
> > > - Data augmentation during training consisted of random cropping and left-right flips, which we believe to be fairly standard for ImageNet training. Preprocessing during training then consisted of resizing the images by 224 x 224 (the code notes that resizing uses bicubic interpolation) and then normalizing with mean and std. During adaptation for ImageNet-C, we simply use TFDS (https://www.tensorflow.org/datasets/catalog/imagenet2012_corrupted), which already resizes the images to 224x224, and we then apply the normalization with mean and std.
> > >
> > > - All methods are loading the exact same model checkpoints. The only differences between methods are in the test time procedure.
> > >
> > > - We confirm that we do compute mCE by first averaging over severities, normalizing by AlexNet scores, and then averaging over corruption types. We also confirm that results are only using the 15 test corruptions, and not the 4 validation corruptions.

---

> > > > ### Comment · Reviewer_EWK6 · 2021-08-30
> > > > **Re: mCE clarifications**
> > > >
> > > > Thanks for clarifying.
> > > >
> > > > To sum up, the numbers I saw so far in our discussion are:
> > > >
> > > > Model | mCE, your ResNetv2 | mCE, ResNet50 reference (torchvision)
> > > > -|-|-
> > > > Baseline | 68.43% (yours)  | 76.7% (e.g. [Schneider et al, Neurips'20](https://arxiv.org/pdf/2006.16971.pdf))
> > > > Batchnorm | 55.79% (yours)  | 62.2% (e.g. [Schneider et al, Neurips'20](https://arxiv.org/pdf/2006.16971.pdf))
> > > > TENT | 53.15% (yours)  | 53.5% (TENT (untuned), [Wang et al, ICLR'21](https://arxiv.org/pdf/2006.10726v2.pdf))
> > > >
> > > > and it seems strange that TENT only improves over batchnorm by ~2% points for your model. Are there any other differences to the original paper in your implementation of TENT? Do you have any way to verify that your implementation will reproduce numbers previously reported in the literature (does not necessary be the papers I am suggesting above) for a known baseline model (e.g. a standard Resnet50)?
> > > >
> > > > To make sure: Does the [TFDS](https://www.tensorflow.org/datasets/catalog/imagenet2012_corrupted) library compute augmentations on the fly from ImageNet images (it seems like that from looking at the docs), or does it take the standardized, already preprocessed (to 224x224px) images distributed by the authors on [zenodo](https://zenodo.org/record/2235448#.YS1kfHUzZhE)? On-the-fly computation will produce numbers not comparable to papers which use the released version of the ImageNet-C dataset.
> > > >
> > > > >  the example script by default uses ResNetv2 (also known as preactivation ResNet, details of which can be found in https://github.com/deepmind/dm-haiku/blob/main/haiku/_src/nets/resnet.py), which we will clarify in our paper
> > > >
> > > > This would be good anyways, thanks.
> > > >
> > > > ---
> > > > EDIT (31/08): I also collected the numbers for the accuracy, and my comment above does not seem to be due to a mismatch in mCE accuracies, but rather in the baseline:
> > > >
> > > > Model | Acc, your ResNetv2 | Acc, ResNet50 reference (torchvision)
> > > > -|-|-
> > > > baseline |            45.64%    |  39.2% [Schneider et al, Neurips'20](https://arxiv.org/pdf/2006.16971.pdf)
> > > > batch norm |         55.97%   |   50.7% [Schneider et al, Neurips'20](https://arxiv.org/pdf/2006.16971.pdf)
> > > > TENT (old results) |	58.05% |     56.0% [Wang et al, ICLR'21](https://openreview.net/pdf?id=uXl3bZLkr3c)
> > > > TENT (tuned)	  |  60.82%    |  ---
> > > >
> > > > So it might be an actual property of the model that the effects of batch norm and TENT are different to that of a standard ResNet50.
> > > >
> > > > Given this insight, I would find it important to check whether metrics are computed on the "true" ImageNet-C dataset (the one checkpointed on Zenodo linked above) and correct the statement about the model in ll. 250-251, "For ImageNet [Deng et al., 2009] experiments, we use the ResNet50" -> it would be good to state that this is a different model from the one typically used (e.g. by the authors of ImageNet-C in their follow-up papers), otherwise readers might confuse the numbers in the same way I did.

---

> > > > > ### Author Response · Authors · 2021-09-01
> > > > > **Regarding Imagenet-C performance**
> > > > >
> > > > > We are not aware of any differences between our implementation of TENT or in our hyperparameters (for untuned TENT) compared to the original paper, after going through both the paper as well as the released code (though the released codebase https://github.com/DequanWang/tent does not include code for reproducing the ImageNet results). It is possible that the different network architecture also leads to different levels of improvement over just batchnorm adaptation, or require slightly different hyperparameters to achieve similar improvements.
> > > > >
> > > > > In our previous post with mCE results, we did not include the tuned results for TENT or BACS (MAP) (we only included the old results and the ensembled results), and it was untuned TENT that achieved an mCE of 53.15. We now include them here for completeness, and note that the improvements of our tuned TENT over BN adaptation appear to be closer to the improvements reported in the original TENT paper (based on the unnormalized error rates reported in the TENT paper).
> > > > > TENT (tuned): 49.76
> > > > > BACS (MAP) (tuned): 48.33.
> > > > >
> > > > > Additionally, in the TENT paper, we also could not find any reported mCE values, nor a table of values of the per corruption error rates from which to compute mCE. Would the reviewer point us to where we can find such results?
> > > > >
> > > > > The TFDS library for ImageNet-C does recompute corrupted images once (after which the inputs are cached, with the corruption procedure being deterministic), and does not simply load the images distributed by the original authors. We will acknowledge in our paper that the results are not necessarily directly comparable to other results using the authors’ originally distributed dataset. We would also be happy to rerun our experiments using the officially distributed datasets for the final version of our paper.
> > > > >
> > > > > We emphasize again that all methods evaluated are using the exact same models and datasets, making the comparison fair regardless of any model or dataset differences from past reported results. While we can rerun our training procedure and evaluation with standard ResNet50 models, it will take a substantial amount of time and effort to do so. We will definitely update our paper to note that we actually use ResNetv2 instead of a standard ResNet.

---

> > > > > > ### Comment · Reviewer_EWK6 · 2021-09-01
> > > > > > **Re: Regarding Imagenet-C performance**
> > > > > >
> > > > > > Dear authors,
> > > > > >
> > > > > > > Additionally, in the TENT paper, we also could not find any reported mCE values, nor a table of values of the per corruption error rates from which to compute mCE. Would the reviewer point us to where we can find such results?
> > > > > >
> > > > > > Thanks for raising this, I am also aware of this issue. The [v2 version on arxiv](https://arxiv.org/abs/2006.10726v2) has these tables in the supplement, and I think that the mCE computed from this would be 53.5% --- however, the TENT paper did not perform hyperparameter tuning, as far as I know. Potentially one of the follow-up papers that cite TENT perform a re-analysis? I used the reference value of 53.5% mCE from the original paper.
> > > > > >
> > > > > > > The TFDS library for ImageNet-C does recompute corrupted images once (after which the inputs are cached, with the corruption procedure being deterministic), and does not simply load the images distributed by the original authors. We will acknowledge in our paper that the results are not necessarily directly comparable to other results using the authors’ originally distributed dataset. We would also be happy to rerun our experiments using the officially distributed datasets for the final version of our paper.
> > > > > >
> > > > > > This is actually an issue in the evaluation, and I think this will be the main reason why the numbers you report are better and do not match the literature (while your updated ImageNet-R numbers do)---most likely this is also . This is also mentioned in the README [in response to an issue reported on Github](https://github.com/hendrycks/robustness/issues/16):
> > > > > >
> > > > > > >> Tiny ImageNet-C has 200 classes with images of size 64x64, while ImageNet-C has all 1000 classes where each image is the standard size. For even quicker experimentation, there is CIFAR-10-C and CIFAR-100-C. Evaluation using the JPEGs above is strongly prefered to computing the corruptions in memory, so that evaluation is deterministic and consistent.
> > > > > >
> > > > > > I think (potentially with the model mismatch) this might explain the performance boost of your baseline model.
> > > > > >
> > > > > > > We emphasize again that all methods evaluated are using the exact same models and datasets, making the comparison fair regardless of any model or dataset differences from past reported results. While we can rerun our training procedure and evaluation with standard ResNet50 models, it will take a substantial amount of time and effort to do so. We will definitely update our paper to note that we actually use ResNetv2 instead of a standard ResNet.
> > > > > >
> > > > > > I agree that the comparisons themselves are fair, and your experimental protocol (after the additional tuning) is thorough.
> > > > > >
> > > > > > My worry is rather that upon publication, your ImageNet-C numbers will be cited anyways without careful double checks. The minimum change you should make to the paper is to outline the potential difference to the standard ResNet50 (if there actually is one, maybe check the reference implementation in torchvision.models with yours), and --- more importantly --- stress that your ImageNet-C numbers are not actually evaluated on the correct test set used in the literature.
> > > > > >
> > > > > > It should e.g. be highlighted in the experimental setup (referencing the library you use), and in the caption of all tables; an idea would be to add a disclaimer "corrupted images were generated on-the-fly, and not taken from the ImageNet-C test set". I am open to other ideas.
> > > > > >
> > > > > > In particular, please check and verify that you do not have any ImageNet-C tables were you compare your numbers to results from the literature (since the test sets are different).

---

> > ### Comment · Reviewer_EWK6 · 2021-08-30
> > **Follow up regarding ImageNet-C performance**
> >
> > Dear authors,
> >
> > first, thanks for all the additional experiments and results and the clear presentation, which make it much easier to interpret the results. In this comment, I will adress your section "Response to ImageNet-C performance".
> >
> > > In table 2, we report accuracies of 65.44 for BACS (ours) compared to 58.05 for TENT, and 59.98 for an ablation of our method BACS (MAP) which does not use ensembling.
> >
> > As a general comment for all tables, I find it important to clearly state which models use ensembling, and which do not use ensembling. I think it is not possible to compare a single TENT model to BACS (ours) because of model size. Ensembling implies a parameter count of 260 million (10x ResNet50 size), which is a lot, and actually comes close to a large EfficientNet-L2 model (480 million parameters) which gets 77.34% top-1 accuracy on ImageNet-C without any adaptation (but with different tricks, I am not calling it a fair comparison --- just want to illustrate that large models make a big difference on ImageNet-C/essentially solve the benchmark).
> >
> > I would propose to make it visually clear in all tables (e.g. like I do below) and plots that the TENT ensemble has to be compared to BACS (ours), and the single TENT to BACS (MAP), and any cross-comparisons do not work simply because the # of parameters (evidently on the order of 5 percent points) have a large confounding effect. Do you agree?
> >
> > > We believe the improvements in accuracy and mCE for BACS (ours) over TENT are substantial.
> >
> > Let me re-group the table you posted above ("Improved results with finer grained tuning") and add the additional information about using ensembling:
> >
> > method | ensemble | Acc (higher is better)|	NLL (lower is better)|	Brier (lower is better)|	ECE (lower is better)
> > -|-|-|-|-|-
> > TENT (tuned)	| no | 60.82|	1.803|	0.522|	0.0313
> > BACS (MAP) (tuned) |no|	61.96|	1.712|	0.5022|	0.0294
> > |||||
> > TENT ensemble (tuned) |yes| 	65.83|	1.586|	0.4726|	0.0956
> > BACS (ours) (tuned) |yes|	66.64|	1.492|	0.4548|	0.0735
> >
> > Firstly, while it is great to see updated results with systematically chosen hyperparameters now, let me note that your original results in Table 2 were misleading in this regard and suggested e.g. an NLL score improvement of 18% from 2.076 -> 1.702! Now it turns out that the actual improvement is from 1.803 -> 1.712 (5%) or 1.586 -> 1.492 (6%), and your response above convinced me that this comes closer to the true value. While revising the paper, I would find it important to avoid overselling the result and make precise quantitative statements about what can be expected from the method (e.g. by communicating percentage improvements that can be expected in the different settings---maybe the relative improvements are even comparable across datasets?). Frankly, I also find a "substantial" improvement much less important than being able to make a correct and realistic estimate of the systematic contribution of your method; I won't make my final score dependent on whether you improve over TENT by 5 or 10% as long as the experiments are sound and as controlled as possible.
> >
> > Secondly, while the mean values are consistently improved, I would find it very important to paint the full picture of variability in these metrics and add a figure in the style of supplement Fig. 4 (the boxplots) to the main paper. The boxplot should be adapted to make it clear which models are comparable (i.e., TENT vs. BACS (MAP) and TENT (ensemble) vs. BACS (ours); e.g. by color, grouping, etc). Another idea would be to plot accuracy for each corruption/severity against the respective score (NLL/Brier/ECE), yielding a scatter plot with a total of 75 points. To avoid this from getting too cluttered, I would limit the plot to TENT vs. BACS and make separate plots for TENT vs. BACS (MAP) and TENT (ensemble) vs. BACS (ours).
> >
> > Such an analysis would be crucial to get a better sense how the metrics are affected by different severities/baseline accuracies. From what I read so far, the result should be that the improvement of BACS is also consistent across severities /"corruption difficulty"? If this is true, this would be a much stronger result than the table showing summary statistics.
> >
> > In general, I would recommend to use more space in the paper for analysing ImageNet-C results, and less for e.g. the MNIST/SVHN domain adaptation experiments, which we already discussed in the other comment.
> >
> > Also, do you plan to "includ[e] results on ImageNet-R for an instance of more "natural" distribution shift." (cf my original review)? Again, it would strengthen the paper if the method would work on another full scale image dataset (using the hyperparameters found on ImageNet-C). ImageNet-R might be a good candidate as it is cheap to run compared to full ImageNet-C.

---

> > > ### Author Response · Authors · 2021-08-30
> > > **Response regarding ImageNet presentation**
> > >
> > > We thank the reviewer again for the helpful suggestions on presentation. We will update our tables to make explicit which methods utilize ensembles and are thus directly comparable and regroup the methods accordingly.
> > >
> > > We will devote more space in our experimental sections to analyze ImageNet results, expanding on the ImageNet-C results by moving the boxplots (and making it visually clear which methods utilize ensembles) from the appendix to the main paper as well as including ImageNet-R results (see table below). To make room for the expanded analysis, we will likely defer the smaller scale CIFAR->STL and SVHN->MNIST results to the appendix. Please let us know if this reorganization addresses your concerns.
> > >
> > > **Imagenet-R results**: We include the ImageNet-R results in the table below using the tuned hyperparameters found for TENT and BACS. We observe that BACS and BACS (MAP) outperform the ensembled and non-ensembled baselines respectively in accuracy, NLL, and Brier score. In terms of ECE, BACS (ours) slightly underperforms compared to TENT ensembles, though still outperforms the vanilla and batchnorm adapted ensemble baselines, as well as all non-ensembled baselines.
> > >
> > > In terms of accuracy, we find BACS (MAP) provides a **17%** improvement over TENT, while BACS (ours) provides a **15%** improvement over TENT ensemble. In terms of error rate, BACS (MAP) provides a **6.5%** reduction compared to TENT, and BACS (ours) provides a **7.1%** reduction compared to TENT ensemble.
> > >
> > > | Method           | Ensemble? | Accuracy | NLL   | Brier  | ECE     |
> > > |------------------|-----------|----------|-------|--------|---------|
> > > | Vanilla          | No        | 24.03    | 5.114 | 0.9245 | 0.1583  |
> > > | BN Adapt         | No        | 25.06    | 4.716 | 0.8848 | 0.07522 |
> > > | TENT             | No        | 27.78    | 4.387 | 0.8406 | 0.05366 |
> > > | BACS (MAP)       | No        | 32.50    | 4.191 | 0.8316 | 0.1110  |
> > > | Vanilla Ensemble | yes       | 26.77    | 4.718 | 0.8752 | 0.08108 |
> > > | BN Ensemble      | yes       | 28.76    | 4.442 | 0.8686 | 0.1344  |
> > > | TENT Ensemble    | yes       | 31.55    | 4.072 | 0.8089 | **0.02411** |
> > > | BACS (ours)      | yes       | **36.40**    | **3.780** | **0.7777** | 0.02684 |

---

> > > > ### Comment · Reviewer_EWK6 · 2021-08-30
> > > > **Re: Response regarding ImageNet presentation**
> > > >
> > > > > To make room for the expanded analysis, we will likely defer the smaller scale CIFAR->STL and SVHN->MNIST results to the appendix. Please let us know if this reorganization addresses your concerns
> > > >
> > > > It does, thanks.
> > > >
> > > > > Imagenet-R results
> > > >
> > > > Thanks for running the additional experiment. However, it adds to the concern I also posted below that something about the baseline numbers might be off. I am not aware of a reference paper using your exact codebase for producing ImageNet-R results, but [Hendrycks et al. (ICCV 2021)](https://arxiv.org/pdf/2006.16241.pdf) report a score of 36.1% top-1 acc for their non-adapted Resnet50 model, and [Schneider et al. (Neurips 2020)](https://arxiv.org/pdf/2006.16971.pdf) report 36.2% top-1 acc for their non-adapted baseline, and 40.1% top-1 acc after batch norm adaptation -- vs. your 36.40% best result after ensembling.
> > > >
> > > > Your numbers seem to have a systematic shift---potentially some issue with re-mapping from 1000 to 200 classes? It would be good to find the cause of the offset. The result looks promising, though.

---

> > > > > ### Author Response · Authors · 2021-09-01
> > > > > **Update on ImageNet-R results**
> > > > >
> > > > > The difference in reported performance on ImageNet-R is indeed due to the relabeling procedure. In the evaluation code of the ImageNet-R repository (https://github.com/hendrycks/imagenet-r/blob/master/eval.py), they restrict the network’s predictions to only the 200 classes present in ImageNet-R. In our previous evaluation, we simply kept the original label space and the network was allowed to make predictions among the 800 classes not included in ImageNet-R, which explains the lower accuracy numbers.
> > > > >
> > > > > We repeat our experiments after restricting the predictions to only the 200 classes present in the dataset, and see improvements that bring our vanilla baseline accuracy similar to previously reported values. We see that BACS (ours) provides the best results across all metrics.
> > > > >
> > > > >
> > > > > | Imagenet-R       | Ensemble? | Acc   | NLL   | Brier  | ECE     |
> > > > > |------------------|-----------|-------|-------|--------|---------|
> > > > > | Vanilla          | no        | 36.40 | 3.288 | 0.7602 | 0.05667 |
> > > > > | BN               | no        | 38.05 | 3.236 | 0.7646 | 0.09744 |
> > > > > | TENT             | no        | 41.13 | 2.967 | 0.7167 | 0.02666 |
> > > > > | BACS (MAP)       | no        | 43.55 | 2.909 | 0.7183 | 0.1074  |
> > > > > | Vanilla Ensemble | yes       | 40.38 | 3.011 | 0.7180 | 0.02339 |
> > > > > | BN Ensemble      | yes       | 42.82 | 3.019 | 0.7408 | 0.1696  |
> > > > > | TENT Ensemble    | yes       | 45.75 | 2.726 | 0.6815 | 0.1025  |
> > > > > | BACS (ours)      | yes       | **47.31** | **2.565** | **0.6625** | **0.01682** |

---

> > > > > > ### Comment · Reviewer_EWK6 · 2021-09-01
> > > > > > **Re: Update on ImageNet-R results**
> > > > > >
> > > > > > Thanks for checking and correcting this. The 200-class protocol is the correct one to use, cf. the paper that introduced the benchmark in the first place. I would include this experiment as a table in the main paper, in the ImageNet section.

---

### Official Review · Reviewer_3Wsi · 2021-07-17

**Rating:** 7
**Confidence:** 4

**Summary:**

A principled (Bayesian) approach to adapt model parameters to potential covariate shift at test time with unlabeled test data. The proposed method does not require to keep training data around for adaptation at inference time, is relatively simple and flexible (initial model training can be carried out with any Bayesian method to learn a posterior density), and provides both high accuracy and strong calibration in experiments.

**Limitations And Societal Impact:**

Well covered overall

**Main Review:**

**Overall appreciation (originality, quality, clarity, and significance)**
- A very well written and executed paper overall
- Language is clear and concise -- very easy to read throughout. Experiments are convincing. Related work is adequately covered. Only a few clarifications needed and typos to correct (see below)
- Closely related to the approach introduced in “Tent: Fully Test-time Adaptation by Entropy Minimization” (Wang et al. 2020) although the proposed algorithm in this paper is derived in a more principled manner, and delivers stronger experimental results in most cases (all experiments except when test data distribution varies drastically from training, eg. SVHN Vs MNIST -- more on this below)

**Clarifying questions**
- Why do you restrict the adaptation procedure to operate without any further access to the original training data? For instance, just keeping the learnt approximate posterior and not the full training data is easier to store and optimize (as you allude to on line 162). Are these any other considerations? What are the potential limitations stemming from this assumption? If relaxed, could it potentially help address the issue you are observing with your method when test data differs drastically from training?
- Your method relies on the availability of several (unlabelled) test points coming from a distribution (potentially) different from training. Did you look into how many such test points are needed for sufficient adaptation across your different experimental settings?
- How do you select hyperparameters (e.g., alpha*)?
- How do you interpret results from Table 2 (e.g., better ECE with a single model and no adaptation)?
- Table 4: is the performance lift over TENT resulting from the ensembling or from the fact you adapt all parameters, not just the batch norm parameters?

**Suggestions**
- Paper is missing a good "figure 1" giving an intuition about / summarizing the approach
- Would add a clarification regarding the availability of unlabeled test points for calibration. This is not always possible to get in certain applications, eg. when test points come in one-by-one
- You are also assuming all test points come from the same test distribution, while in practice they may be coming from different distributions (e.g., evolution of underlying dynamics over time)
- Line 192: would clarify here that TENT focuses on BN parameters (instead of lines 369-371) -- that would help better understanding experimental results

**Minor points**
- Line 34: typo “minmization’
- Lines 145-147: you may be missing a word in this sentence
- Line 147: drop “that” after “However”
- Line 341: missing a “the” after “We also find that”
- Line 396: need to replace BENT by BACS?
- Line 398: drop duplicate word “objectives”


**Time Spent Reviewing:**

7

---

> ### Author Response · Authors · 2021-08-10
> **Response to Reviewer 3Wsi**
>
> **Motivations for test-time adaptation**: The primary reason we restrict we operate without access to training data is for reduced storage and computational efficiency. Another motivation for test time adaptation could be for privacy reasons. For example, we may not wish to share individual patient data from training hospitals when adapting a new model at a new hospital. The main limitation of this assumption in our approach is that we are now constrained by a learned approximate posterior, which may be inaccurate. We do believe that if this assumption were relaxed to allow some or all of the training data during adaptation, this would allow us to avoid the overly limiting constraint of the approximate posterior on test sets that differ drastically from the training set.
>
> **Hyperparameter selection**: Please see the reply addressed to all reviewers for a discussion of hyperparameter selection and improved results upon more fine-grained tuning of hyperparameters on the ImageNet-C validation corruptions.
>
> **Benefits of BACS-posterior over TENT in SVHN->MNIST**: The improvement over TENT is primarily due to ensembling. For SVHN->MNIST we find that ensembles of TENT models perform similarly to BACS - posterior. We also included experiemnts in the reply to all reviewers evaluating TENT ensembles on ImageNet-C, where we find that BACS (ours) outperforms TENT ensembles.

---

> > ### Comment · Reviewer_3Wsi · 2021-08-31
> > **Thank you for the response**
> >
> > Dear authors,
> >
> > Thank you for the response and additional analyses.
> >
> > Q1 -- Thanks for clarifying. Would suggest including this explanation in the text (e.g. as a replacement for the explanation lines 28-30 which I relatively find weak in comparison). Also the second argument on data privacy makes more compelling/impactful the several analyses carried out on smaller scale datasets (while just considering the first argument on memory constraint may lead to the conclusion that analyses on smaller datasets are not that important).
> >
> > Q2 -- I could not find an answer to this question.
> >
> > Q3 -- Thank you for the additional analyses. Would highly encourage you to include these in the revised manuscript, including hyperparameter search for TENT since these were missing from the original paper.
> >
> > Q5 -- Thank you for clarifying. I think it would be interesting to call that out in the text since the ensembling stems from your Bayesian perspective and therefore is a key strength of your method.
> >
> > Regarding the comparison with DA baselines suggested by other reviewers -- I think this is a fair point. In my view this could be addressed by a) explaining more clearly in the introduction your particular setting of interest and why these methods are not relevant (eg. ability to work offline and online, no need to train generative models, etc) b) caveating in the related work section that in other contexts, the DA methods (referenced by reviewers EWK6 and 6Ge4) may perform better and discuss high level pros/cons Vs these methods.

---

### Author Response · Authors · 2021-08-10
**Shared response to all reviewers regarding hyperparameter tuning**

We thank all the reviewers for the helpful feedback. As multiple reviewers asked about how hyperparameters were tuned, we include this discussion here addressed to all reviewers. Please also see the responses to each individual reviewer for other questions.

Our method introduces two additional hyperparameters compared to TENT: the ensemble size and the weighting term $\beta = (1/\alpha)$ that controls how to weight the training posterior against the entropy on the test inputs. We used the validation corruptions in CIFAR and Imagenet Corrupted to coarsely tune and validate hyperparameters, though we opted to do very minimal tuning to reflect that we often may not have access to validation sets that accurately reflect the distribution shifts we may encounter.

**Tuning procedure for beta**: The weighting coefficient beta was the only hyperparameter we tuned (very coarsely). We heuristically decided on an initial value of $\beta = 0.0001$ (with a test dataset of 10000 test inputs, this gives the same weight to the losses for each individual labeled and unlabeled data point), and evaluated BACS on the validation corruptions of CIFAR10-C and CIFAR100-C. We observed the performance of BACS was sensible on these validation corruptions, and so we fixed this value for all the small scale experiments we report with ResNet26.

On ImageNet-C, we again tried the same value of $\beta=0.0001$ on the validation corruptions, and observed that performance was worse than the simple batchnorm adapted baseline, since accuracy on some of the corruptions dropped to almost 0 after adaptation. We then increased $\beta$ by a factor of 10 (to our final value of 0.001), observed improved performance over TENT and BN adaptation on the validation corruptions, and then used this value of $\beta$ to collect our results on the full ImageNet-C datasets.

For ensemble size, we simply kept it fixed at 10 ensembles, using the same ensemble size used in Nado et al, which evaluated ensembles with adapted batch norm statistics..

**Number of epochs**: A hyperparameter we share with TENT is the number of epochs. We will emphasize in the text that a default value of 1 epoch (as recommended by TENT) is already sufficient to provide substantial improvements (and outperform baselines) across our experiments.

On SVHN->MNIST (where the distribution shift is more drastic), we evaluate with 10 epochs as well to match the experiments in the TENT paper. On CIFAR10/100-C, we additionally evaluate results with 5 epochs of adaptation in order to illustrate that our method can benefit from additional computation (improving both accuracy and uncertainty estimation). On the other hand, TENT, with unconstrained entropy minimization and without ensembling, becomes excessively overconfident as more training occurs and we can observe worse performance in terms of the uncertainty aware metrics like NLL.

**Other adaptation hyperparameters**: Additional hyperparameters shared with TENT are the learning rate and batch size during adaptation. We simply kept fixed at the values taken from the TENT paper, as they also evaluated using CIFAR10/100-C, ImageNet-C, and SVHN->MNIST. We did not tune any hyperparameters for TENT or our method for our experiments.

**Improved results with finer grained tuning**: We now include results after conducting a more fine grained hyperparameter search using the ImageNet-C validation corruptions. For TENT, we tune the learning rate, while for BACS, we jointly tune the learning rate and beta.

The ranges of values considered were

TENT: Learning rate: [0.0025, 0.001, 0.0005, 0.00025 (previously used value), 0.0001]

BACS: $\beta$: [0.003, 0.001 (previously used value), 0.0003], learning rate = [0.005, 0.00025 (previously used value), 0.0001, 0.00005].

We found that TENT can improve with increased learning rate, with the final value selected as 0.001 (compared to 0.00025 as taken from the TENT paper). For BACS, we are also able to improve performance by lowering beta and the learning rate slightly, with final values chosen as beta=0.0003, and learning rate= 0.0001.

We now report results of the tuned methods (bottom 4 rows) across the main ImageNet-C test sets. As suggested by reviewer EWK6, we also include an ensemble of  models adapted via TENT as a baseline.  We see that after additional tuning, BACS provides the best results across accuracy, NLL, and Brier score, with the non-ensembled ablation BACS (MAP) providing the best results overall in ECE.

|                               | Acc (higher is better) | NLL (lower  is better) | Brier (lower is better) | ECE (lower is better) |
|-------------------------------|------------------------|------------------------|-------------------------|-----------------------|
| TENT (old results)            | 58.05                  | 2.076                  | 0.5529                  | 0.08137               |
| TENT (ensemble) (old results) | 63.77                  | 1.857                  | 0.5218                  | 0.1655                |
| BACS (MAP) (old results)      | 59.98                  | 1.927                  | 0.5293                  | 0.06642               |
| BACS (ours) (old results)     | 65.44                  | 1.702                  | 0.4542                  | 0.1479                |
| TENT (tuned)                  | 60.82                  | 1.803                  | 0.522                   | 0.0313                |
| TENT ensemble (tuned)         | 65.83                  | 1.586                  | 0.4726                  | 0.0956                |
| BACS (MAP) (tuned)            | 61.96                  | 1.712                  | 0.5022                  | **0.0294**                |
| BACS (ours) (tuned)           | **66.64**                  | **1.492**                  | **0.4548**                  | 0.0735                |


[1] Z. Nado, S. Padhy, D. Sculley, A. D’Amour, B. Lakshminarayanan, and J. Snoek. Evaluating Prediction-Time Batch Normalization for Robustness under Covariate Shift. arXiv, 6 2020. URL https://arxiv.org/abs/2006.10963.

---

> ### Comment · Reviewer_EWK6 · 2021-09-01
> **ImageNet-C Results**
>
> Dear authors, dear reviewers,
>
> as I noted in my comments below (a bit hidden in the long question/answer thread with the authors), while most of my other comments have been adequately adressed, the paper currently has an inconsistency in the ImageNet-C evaluation.
>
> Namely, all reported ImageNet-C numbers are generated on corrupted images the authors generated themselves, not on the official release of the ImageNet-C test set. I find it therefore important that this is properly highlighted in the paper, to prevent the reported numbers from appearing in follow-up work in the wrong context.
>
> Please also make sure that all tables on ImageNet-C exclusively contain numbers you produced yourself on your custom version of the dataset. It is impossible to compare the numbers on the "in memory" ImageNet-C to existing numbers in the literature produced on the real test set, like e.g. [Ford et al (2019)](https://arxiv.org/pdf/1901.10513.pdf) note:
>
> > The publicly released Imagenet-C dataset as .jpeg files is significantly harder than the same dataset when the corruptions are applied in memory. It appears that this is due to additional artifacts added to the image from the JPEG compression algorithm (see Figure 8). Future work should make care of this distinction when comparing the performance of their methods, in particular we note that the results in (Geirhos et al., 2018; Hendrycks & Dietterich, 2018) were both evaluated on the jpeg files.
>
> [Ford et al (2019)](https://arxiv.org/pdf/1901.10513.pdf) include an ablation study on this in their paper, and report the results in Table 1 in their appendix. *The difference is in the range of about 5 to 10 percent points*, which is consistent with the mismatch in baseline accuracy the authors report here.
>
> Summing up, I am convinced that the evaluation the authors performed is thorough and insightful, and their proposed method works. The rebuttal phase added many important results (better baselines, ImageNet-R results). If the authors agree to address the concern about ImageNet-C in the following way (open for alternative suggestions, but this sounds like a reasonable strategy to me):
>
> - Cite [Ford et al (2019)](https://arxiv.org/pdf/1901.10513.pdf) prominently and put a disclaimer in their main text that ImageNet-C numbers are not produced on the official test set, and are therefore better by the mentioned offset.
> - Make this difference clear in any table or figure with ImageNet-C results (I would recommend to give the protocol a name, like "ImageNet-C (in memory corruptions)") and explain this once in the main text, then it will take a minimum amount of space.
> - Include a full run on the real ImageNet-C test set with the common evaluation protocol for the camera ready release (I am fine with the fact that the hyperparameter tuning was performed on the custom dataset if this is mentioned), to make the results comparable to the literature.
>
> along with the changes promised in the comments below (e.g. regarding the presentation of results, inclusion of the ensemble TENT baseline, domain adaptation, etc), I am willing to update my score to (6).

---

> > ### Author Response · Authors · 2021-09-01
> > **Agree to address ImageNet-C discrepancies**
> >
> > We thank the reviewer again for all helpful comments!
> >
> > We confirm that all our ImageNet-C results were produced with the exact same setup and will make sure all tables make it clear what dataset was being used. We will cite Ford et al in our experiments section with disclaimer about the differences between the version of ImageNet-C we use and the official release. We will also include full results with the official released dataset as well, and will also incorporate the previously promised changes.

---

> > > ### Comment · Reviewer_EWK6 · 2021-09-01
> > > **Re: Agree to address ImageNet-C discrepancies**
> > >
> > > Dear authors,
> > >
> > > thanks a lot, this sounds great.  Please see my post-rebuttal comments which I added to my original review. I decided to increase my score to (7).

---

### Decision · Program_Chairs · 2021-09-27

**Decision:**

Accept (Poster)

**Comment:**

This paper proposes a Bayesian approach to covariate shift using unlabelled test data. There was extensive discussion of the paper, and number of concerns, many of which have moved towards a resolution throughout the response period. Key issues included:
1. Use the standard ImageNet-C test set, instead of the custom one in the submission.
2. Hyperparameter tuning
3. Show results for ensembles of baselines
4. Missing comparisons of DA baselines
5. Concerns about small-scale datasets
6. mCE computation inconsistencies

While it is concurrent work, and does not factor into the evaluation of this submission, it could also be useful to the readership to discuss the recent paper (https://arxiv.org/abs/2106.11905), which shows there are in fact risks of Bayesian model averaging under covariate shift, and explains why deep ensembles don't suffer from these risks.

Overall, this is nice work, and I am supportive of this paper. Please do carefully incorporate all reviewer comments in finalizing the camera-ready, especially point 1, where it is crucial to show an evaluation using the standard test-set for ImageNet-C.